# The core and accessory Hfq interactomes across *Pseudomonas aeruginosa* lineages

Julian Trouillon [1,3,4], Kook Han[2,4], Ina Attrée [1] & Stephen Lory [2✉]

The major RNA-binding protein Hfq interacts with mRNAs, either alone or together with regulatory small noncoding RNAs (sRNAs), affecting mRNA translation and degradation in bacteria. However, studies tend to focus on single reference strains and assume that the findings may apply to the entire species, despite the important intra-species genetic diversity known to exist. Here, we use RIP-seq to identify Hfq-interacting RNAs in three strains representing the major phylogenetic lineages of *Pseudomonas aeruginosa*. We find that most interactions are in fact not conserved among the different strains. We identify growth phase-specific and strain-specific Hfq targets, including previously undescribed sRNAs. Strain-specific interactions are due to different accessory gene sets, RNA abundances, or potential context- or sequence- dependent regulatory mechanisms. The accessory Hfq interactome includes most mRNAs encoding Type III Secretion System (T3SS) components and secreted toxins in two strains, as well as a cluster of CRISPR guide RNAs in one strain. Conserved Hfq targets include the global virulence regulator Vfr and metabolic pathways involved in the transition from fast to slow growth. Furthermore, we use rGRIL-seq to show that RhlS, a quorum sensing sRNA, activates Vfr translation, thus revealing a link between quorum sensing and virulence regulation. Overall, our work highlights the important intra-species diversity in post-transcriptional regulatory networks in *Pseudomonas aeruginosa*.

[1] Université Grenoble Alpes, CNRS, CEA, IBS UMR 5075, 38044 Grenoble, France. [2] Department of Microbiology, Harvard Medical School, Boston, MA, USA. [3] Present address: Institute of Molecular Systems Biology, ETH Zurich, Zurich, Switzerland. [4] These authors contributed equally: Julian Trouillon, Kook Han. ✉email: stephen_lory@hms.harvard.edu

Post-transcriptional regulation is an important regulatory mechanism allowing bacteria to control protein synthesis in response to rapid environmental changes. The stability or accessibility of a messenger RNA (mRNA) is often modified through interactions with RNA-binding protein (RBPs) alone or in complex with a small noncoding RNA (sRNA)[1]. The recent advances in deep sequencing resulted in the discovery of thousands of sRNAs suggesting that bacterial post-transcriptional regulatory networks are far more extensive than previously thought[2]. Widely distributed across the bacterial kingdom, Hfq is one of the major regulatory RBPs that has a prime importance for the regulation of key biological processes[3]. The most extensively studied functions of Hfq are its roles in post-transcriptional regulation through various mechanisms[4]. Hfq can notably facilitate the base-pairing interactions between regulatory sRNAs and their mRNA targets[5] or act as an sRNA-independent translational repressor[6]. Recent findings also suggest that Hfq plays a role in ribosomal rRNA processing[7,8] and interacts with mRNAs that likely are not regulated by sRNAs[9]. In the opportunistic pathogen *Pseudomonas aeruginosa*, Hfq was shown to be involved in the regulation of diverse functions ranging from the synthesis of virulence factors[10,11], carbon metabolism[12], and antibiotic resistance[13], to a switch between planktonic and biofilm lifestyles[14].

While most of the current knowledge on bacterial regulatory networks usually derives from work on a single reference strain and is assumed to apply to the entire species, we are only starting to grasp the vast intraspecies genetic diversity that can be found in some bacteria. This raises the question of co-evolution of regulatory elements and their targets, in particular in organisms with mosaic genomes that have evolved through horizontal gene transfer. *P. aeruginosa* embodies that fact with now more than 5000 sequenced genomes spanning various intraspecies lineages with specific characteristics[15]. Numerous studies have focused on the genomic characterization and identification of these different lineages and found various important differences in most central cellular processes[15–18]. Indeed, *P. aeruginosa* core genes represent only about 1% of the entire species pan-genome, which explains large differences in antimicrobial and virulence-related phenotypes[15]. For example, strains belonging to three different *P. aeruginosa* lineages harbor mutually exclusive virulence-related secretion systems[19,20] and carry different antibiotic resistance determinants[21]. Additionally, we recently showed that transcription factors of the response regulator family exhibit large functional variability across these three *P. aeruginosa* lineages[22]. While the discovery of these differences calls for further molecular investigation, very few studies have focused on identifying potential intraspecies regulatory network specificities.

Here, we determined the growth phase-dependent as well as strain-dependent RNA interactomes of Hfq in different *P. aeruginosa* strains, each selected from the three representative lineages, by RNA co-immunoprecipitation and sequencing (RIP-seq). We describe the core and accessory interactomes of Hfq and highlight important targeted cellular mechanisms. Our approach identified two new major strain-dependent Hfq targets; most of the operons of the Type III Secretion System (T3SS) apparatus and secreted effectors in two lineages, and a cluster of CRISPR RNAs in the third. Additionally, we found that the mRNA of the global regulator Vfr interacts with Hfq in all three strains and further identified the involved sRNA using rGRIL-seq[23]. The RhlS sRNA, previously studied for its role in quorum sensing, was found to activate *vfr* transcription, revealing a new link between quorum sensing and virulence regulation. The untargeted analysis of the obtained datasets also resulted in the discovery of potential new sRNAs across the three lineages. Altogether, our comparative study reveals a wide functional intraspecies diversity of Hfq interactions as well as important conserved targets.

## Results

**Comparison of RBPs and sRNAs across *P. aeruginosa* lineages.** The phylogenetic analysis of core genes in 192 complete *P. aeruginosa* genomes revealed three distinct major lineages (Fig. 1a) in accordance with previous reports[15]. The two most populated groups are represented by reference laboratory strains PAO1 and PA14[24,25]. The third group, recently delineated into two subgroups[15], includes its first fully sequenced member, PA7[21], as well as IHMA879472[26] (IHMA87), a strain characterized for its Exolysin A-dependent hypervirulence[27]. At the whole-genome level, the 3rd group of strains share a significantly lower average nucleotide identity (ANI) (~93–98%) with the two other groups and among other differences in gene content, they are devoid of T3SS and cognate secreted effectors. While the three lineages differ greatly in their accessory genetic content, the major regulatory RBPs Hfq, RsmN, and RsmA, known for their role in controlling various important processes, are conserved between all *P. aeruginosa* strains (Fig. 1b), raising the question of the functional conservation of their cognate regulatory sRNAs and target mRNAs. Hfq was found particularly well conserved, with 100% of sequence identity between PAO1, PA14, and IHMA87. Interestingly, ProQ, originally identified as a major sRNA-binding regulator in *Salmonella enterica*[28], exhibits a five amino acids insertion at the N-terminus in the PA7/IHMA87-like lineage (Fig. 1b). To further investigate potential differences in post-transcriptional regulatory networks, we examined the conservation of 200 sRNAs previously identified in PA14[29] in all 192 complete genomes of *P. aeruginosa*[30] (Fig. 1c). Strikingly, only six strains phylogenetically very close to PA14 possess more than 90% of these sRNAs, ~70% of them were detected across the PA14 and PAO1 groups, and only half is present in the 3rd group, giving a first glance at the diversity of sRNAs found between and within *P. aeruginosa* groups of strains. Therefore, the sRNA genomic content is strain-dependent and reveals the need for interstrain comparison studies.

**The Hfq interactome in *P. aeruginosa* lineages during growth.** In the light of the observed differences in predicted sRNA content, we investigated the interactome of the major regulatory RBP Hfq in the representative strains from each *P. aeruginosa* lineage (Fig. 1a). To that aim, we chromosomally inserted a C-terminal 3xFLAG tag into *hfq* in *P. aeruginosa* PAO1, PA14, and IHMA87. We demonstrated that the expression of Hfq-3xFLAG did not affect the growth of the corresponding *P. aeruginosa* strain, and that they were functional as tagged Hfq immunoprecipitated PrrF1 as expected, an sRNA conserved in all three strains and known to bind Hfq (Supplementary Fig. 1). Using Hfq-3xFLAG expressing strains of PAO1, PA14, and IHMA87, we performed RNA-co-Immunoprecipitation followed by deep sequencing[31] (RIP-seq) at exponential, transition and stationary growth phases, and analyzed enriched RNAs compared to control samples from cells expressing untagged Hfq (Fig. 2a–c; Supplementary Data 1; Supplementary Fig. 2). Similar patterns of growth phase clustering were observed for the three strains. Namely, (i) a 30–50% core set of targets was enriched in all three growth phases, (ii) the transition and stationary phases seem to be more similar to each other than to exponential phase, and (iii) specific sets of targets were identified in each growth phase, with a higher number of specific targets during stationary phase. Functional enrichment analysis allowed the identification of major biological processes enriched in all nine combinations (strains and growth phases). Notably, we found that the functional group containing mRNAs for branched-chain amino acids (BCAA) degradation enzymes was enriched in a growth phase-dependent manner (Fig. 2d), in concordance with their known regulatory role in response to amino acid starvation

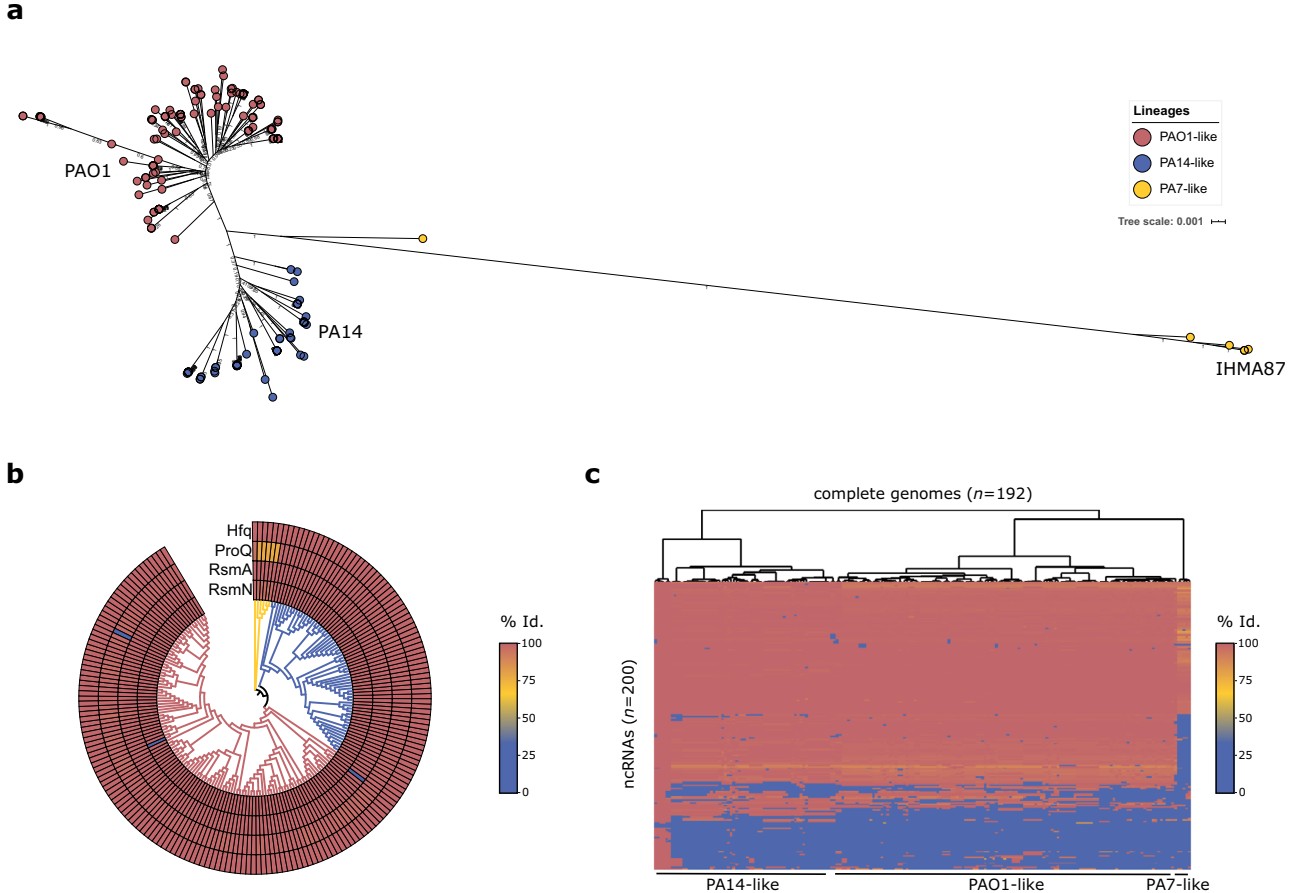

**Fig. 1 Conservation of *P. aeruginosa* post-transcriptional regulatory networks. a** Maximum-likelihood phylogenetic tree of 192 *P. aeruginosa* complete genomes. The tree was generated from the multiple alignment of the concatenated sequences of 66 core genes for each strain with 100 bootstraps. PAO1-like strains are depicted in red, PA14-like in blue and PA7/IHMA87-like in yellow. **b** Heatmap showing % protein sequence identity with *P. aeruginosa* PA14 sequences for four major regulatory RBPs. Strains are arranged on the rounded phylogenetic tree from **a** with leafs colors corresponding to the three different strain lineages. **c** Hierarchical clustering analysis of the conservation of 200 sRNAs in the 192 *P. aeruginosa* genomes. The clustering was performed on both rows and columns with generation of a dendogram based on Manhattan distances for strains. The heatmap shows % DNA sequence identity with PA14 sRNA sequences. Source data are provided as a Source Data file.

through the DNA-binding protein Lrp[32]. This result reveals a potential new conserved role for Hfq in the modulation of BCAA-dependent growth inhibition, as it was already suggested in *Vibrio alginolyticus*[33]. Transcripts for other growth-dependent metabolic pathways, such as the synthesis and degradation of ketone bodies, were enriched in the late phases of growth, further supporting the known central role of Hfq in growth rate determination and starvation response. One such example is *coxB*, encoding a subunit of the $aa_3$ cytochrome c oxidase, a major component of the aerobic respiration machinery[34]. The *coxB* RNA was bound to Hfq across all growth phases and strains (Fig. 2g), which might contribute to the known, unexplained repression of the *cox* genes during growth in rich media[35]. Similarly, two-component systems (TCSs) and bacterial chemotaxis related mRNA targets were also enriched in the later phases of growth (Fig. 2e; Supplementary Data 2), in agreement with the known role of Hfq in community-based behaviors displayed in high cell density conditions[36]. Indeed, numerous transcripts encoding major transcription factors were found enriched. Notably, the *erbR* RNA, encoding a TCS response regulator involved in the regulation of the EraSR TCS, the cytochrome $c_{550}$ and of pyrroloquinoline quinone (PQQ) biosynthesis[37], was bound to Hfq in all growth phases and strains (Fig. 2h). Additionally, the *antR* RNA was highly enriched in specific growth phases and strains (Fig. 2i). AntR is a transcriptional activator of genes involved in the degradation of

anthranilate, a precursor of several 2-alkyl-4(1H)-quinolone metabolites (AQs), including the quorum-sensing molecules PQS and HHQ[38]. In PAO1, it was shown that the PrrF1 and PrrF2 sRNAs interact with the *antR* RNA, which interaction is facilitated by Hfq, to inhibit *antR* translation, leading to increased quorum-sensing signaling[39]. Our results suggest that this interaction happens in both PAO1 and IHMA87, predominantly during the transition phase, which accordingly correspond to the expected peak activation of quorum-sensing signaling. No enrichment of *antR* was detected in PA14, which matches previous results where the *antR* mRNA was found to have a shorter 5' leader sequence in this strain, leading to the loss of the Hfq-binding site on the transcript[14,39]. Finally, transcripts for genes involved in carbon metabolism and TCA cycle, were enriched during growth (Fig. 2f), which corroborates the well-established function of Hfq in central metabolism regulation through binding to the catabolite repression control protein Crc[9,40]. Altogether, our approach clearly delineates the growth-dependent interactomes of Hfq and show that it might globally affect the core and accessory metabolisms as well as bacterial communication systems. The Hfq-dependent interactions with components of these key mechanisms seems overall well conserved between the three tested strains.

The global comparison revealed 427 common Hfq targets between the three strains. Strain-specific targets were also

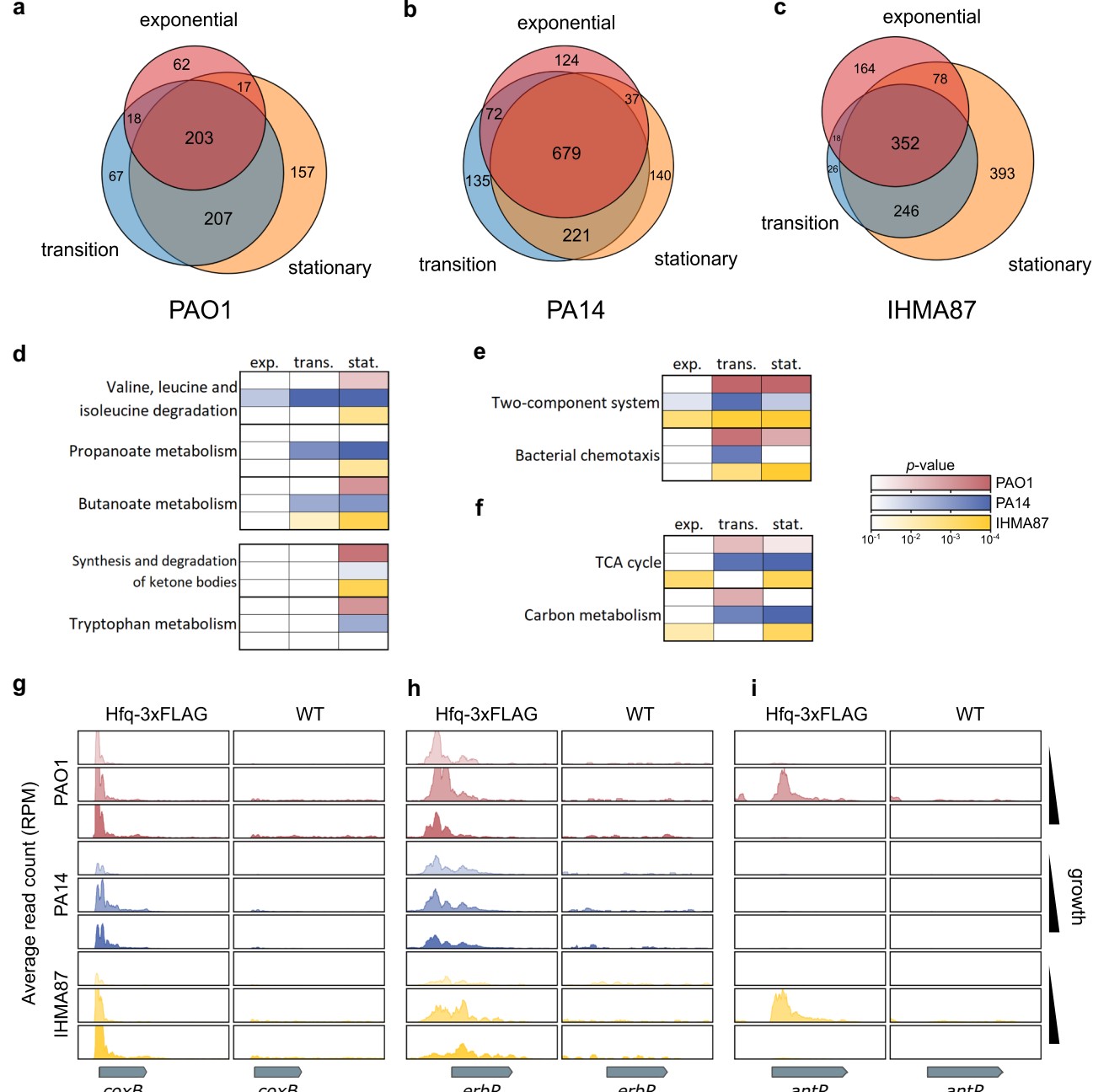

**Fig. 2 Determination of Hfq interactome during bacterial growth. a–c** Venn diagram showing the comparative analysis of number of RNA targets between growth phases for PAO1 (**a**), PA14 (**b**), and IHMA87 (**c**). **d–f** Heatmaps showing results of functional enrichment analysis for selected KEGG pathways and GO Terms for each strain and growth phase using the DAVID statistical analysis[85]. Only genes shared with PAO1 homologs were used. **g–i** Read coverages, averaged between biological replicates (n = 2), at the *coxB* (**g**), *erbR* (**h**) and *antR* (**i**) loci for Hfq-tagged and control samples in all strains and growth phases. Source data are provided as a Source Data file.

identified: 133 for PAO1, 562 for PA14 and 542 for IHMA87 (Fig. 3a). Among the 427 common targets, 109 were found associated with Hfq in all three samplings of growth stages (Fig. 3b). These 427 growth-dependent and 109 growth-independent core *P. aeruginosa* targets of Hfq, span major key cellular functions (Fig. 3c). The set of growth-independent core targets includes two-component systems and transcription regulators, demonstrating the important role of Hfq as a master regulator of transcriptional as well as post-transcriptional regulatory networks. Interestingly, among strain-specific Hfq targets, only ~30% of them are transcribed

from the accessory genome (Fig. 3d), while the rest are conserved in at least two strains, suggesting that most of the differences between strains arises from differential binding of Hfq to the same conserved target. Among the strain-specific targets, there were few mRNAs encoding enzymes of secondary metabolic pathways (Fig. 3e), which could potentially correspond to the specific individual niches of these strains. Notable among these targets is *hfq* mRNA, paralleling the observation that it is subject to autoregulation in *E. coli*[41]. The global virulence factor regulator *vfr*, which mRNA was previously reported to interact with both the Hfq and RsmA RNA-binding

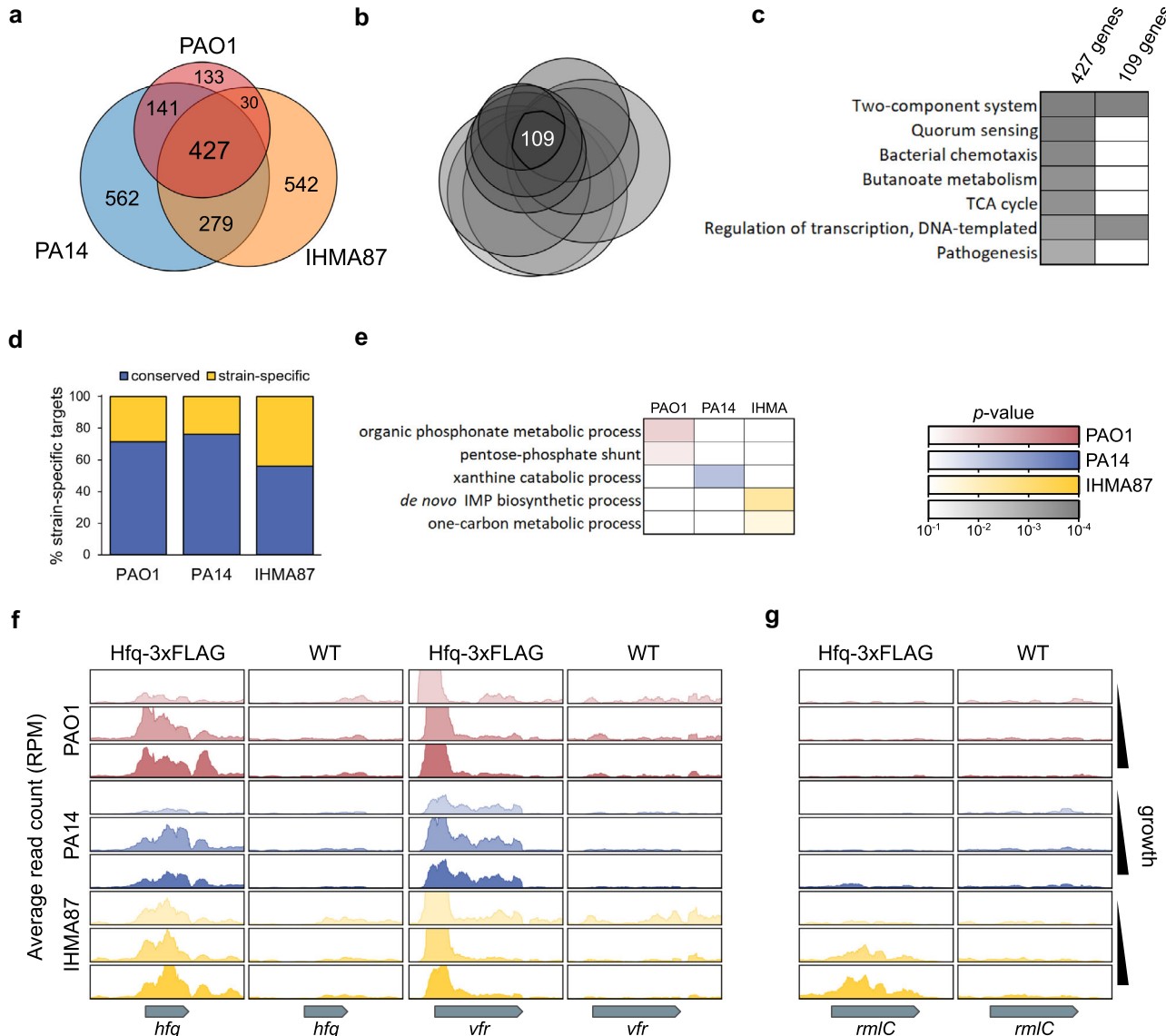

**Fig. 3 Comparative analysis of Hfq targets across strains. a** Venn diagram showing overall Hfq target overlaps between all three strains. **b** Venn diagram showing Hfq targets overlaps between all combination of strains and growth phases. **c** Results of functional enrichment analysis of core Hfq targets using the group of 427 growth-dependent core targets identified in **a** and the group of 109 growth-independent core targets identified in **b**. Only genes shared with PAO1 homologs were used on the DAVID website[85]. **d** Proportion of strain-specific Hfq targets that are shared between genomes (blue) or that are genes specific to the corresponding strain (yellow). **e** Results of functional enrichment analysis of strain-specific Hfq targets from DAVID analysis. **f** Read coverages, averaged between biological replicates ($n = 2$), at the *hfq* and *vfr* genes for Hfq-3xFLAG tagged and control samples (Hfq lacking the FLAG tag) in all strains and growth phases. **g** Same as in **f** for the *rmlC* gene. Read counts are expressed in read per million reads (RPM). Source data are provided as a Source Data file.

proteins[42,43], was also strongly enriched in our Hfq RIP-seq data in all strains and growth phases (Fig. 3f), as further explored below. Interestingly, one of the accessory targets identified in IHMA87 was *rmlC*, encoding an isomerase responsible for core oligosaccharide and O polysaccharide assembly[44] (Fig. 3g). This result suggests a role of Hfq in regulation of LPS synthesis in IHMA87 which is O11_O12 serotype[27], and not in PAO1 or PA14 which belong to serotypes O5 and O10, respectively.

**Annotated and novel Hfq-bound sRNAs.** To account for all Hfq-bound sRNAs, we integrated in the analysis 200 sRNAs from PA14[29] and the corresponding 150 and 101 homologs identified in silico in PAO1 and IHMA87, respectively (Supplementary Data 3). We found ~40% bound to Hfq in at least one growth

phase in each strain (Fig. 4a), highlighting again the pivotal role of Hfq in sRNA-driven post-transcriptional regulation. Interestingly, most of the differences between strains were due to differences in conservation of sRNAs themselves and not different binding to Hfq as the three strains displayed a globally similar pattern of sRNA enrichment. This suggests that sRNA-Hfq interactions are relatively conserved, and that strain-specific sRNA-Hfq interactions mainly evolved from different genomic sRNA pools. Most previously studied sRNA-Hfq interactions were conserved across the strains, such as the interaction with PrrF1 and PrrF2 which was found increasingly more pronounced with growth (Fig. 4b). Surprisingly, the major regulatory sRNA RsmY was found significantly enriched in PAO1 and IHMA87, but not in PA14 (Fig. 4c). RsmY seemed slightly less abundant in PA14 in late growth phases, as assessed in RIP-seq control

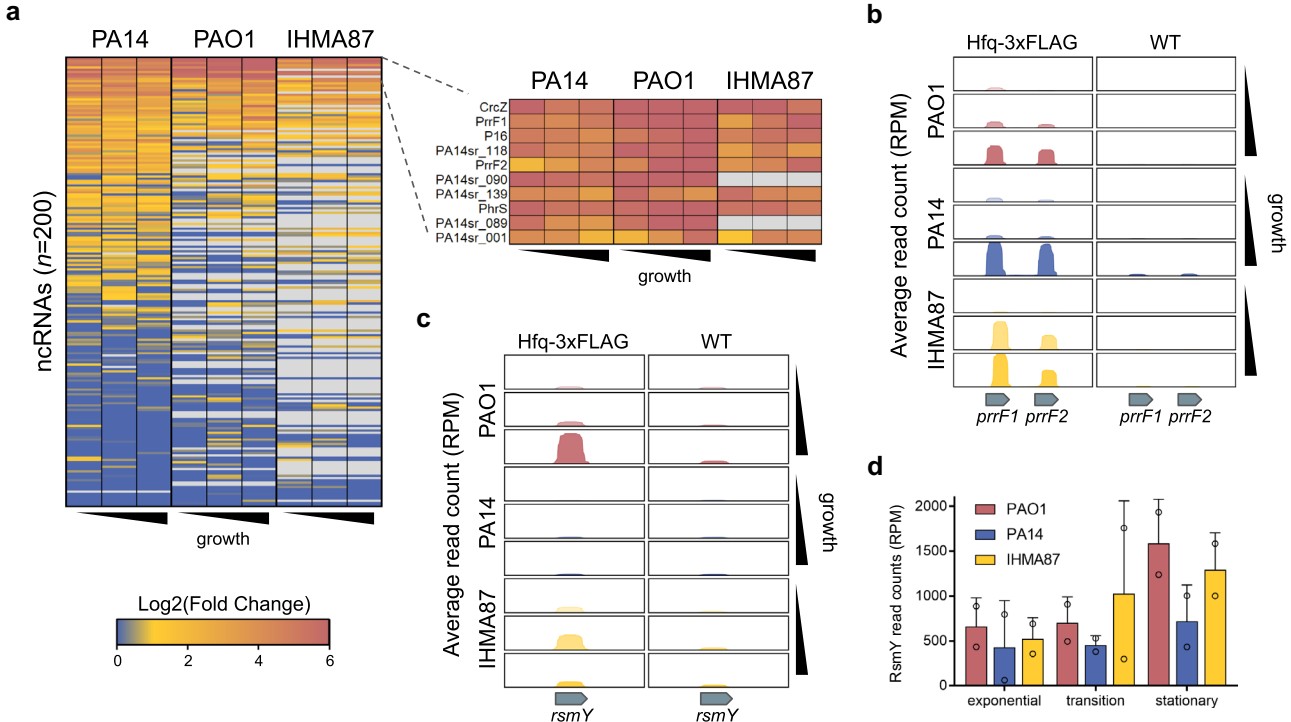

**Fig. 4 sRNAs interactions with Hfq. a** Heatmap of fold enrichments in Hfq-3xFLAG samples compared to WT for a set of 200 sRNAs in all strains and growth phases. **b, c** Read coverages, averaged between biological replicates ($n = 2$), at the *prrF1-2* (**b**) and *rsmY* (**c**) loci for Hfq-tagged and control samples in all strains and growth phases. **d** Normalized read counts for the sRNA RsmY in RIP-seq control samples. Read counts are expressed in read per million reads (RPM) as mean ± SD for two biological replicates. Source data are provided as a Source Data file.

samples (Fig. 4d), which may be due to a mutation in the upstream regulator histidine kinase LadS in PA14[45] and could explain the difference in RIP-seq enrichment.

Most specificities in sRNA-Hfq interactions were due to differences in sRNAs content, thus raising the need of identifying all strain-specific sRNAs. To that aim, we performed an annotation-independent peak-calling analysis in order to rescue missed targets and unannotated sRNAs using the CLIP-seq and RIP-seq peak caller PEAKachu (https://github.com/tbischler/PEAKachu) (Supplementary Data 4). A total of 332 strictly intergenic peaks was identified across all strains and growth phases (Supplementary Data 5). Among these, we identified seven sRNAs predicted structures matching the StructRNAfinder bacterial sRNA database[46] (Supplementary Fig. 3 and Supplementary Data 6), which could correspond to unannotated sRNAs from diverse known families of sRNA. Two of them were actually the same sRNA detected in both PA14 and IHMA87 genomes, and its sequence can also be found in PAO1. In PA14, this sRNA falls exactly after a previously identified TSS that was not associated with any transcript but proposed to correspond to an unidentified intergenic sRNA[29]. Moreover, more than half of intergenic peaks showed high enrichment (>10-fold), suggesting that, even if not matching sRNA databases, more new regulatory sRNAs might be present in these regions. Also, antisense intragenic peaks were identified in 286 genes across strains (Supplementary Data 7), revealing a wide array of potential antisense regulatory sRNAs, which are common across the bacterial kingdom and often regulators of their cognate gene[47]. Notably, a new antisense transcript, encompassing a predicted sRNA secondary structure close to the *Mycobacterium* sRNA Ms_IGR-4, was detected in PAO1 and PA14, transcribed within the *dnaA* gene, which encodes a replication initiation protein essential for bacterial viability[48] (Supplementary Fig. 4). The detected transcript is found directly downstream of a transcription start site that was previously detected in PA14 but not associated with any transcript[29]. The binding of Hfq to a *dnaA*-antisense sRNA reveals a potential new regulatory mechanism for a key cellular process; i.e., DNA replication, that calls for further investigation. Overall, the untargeted analysis of our RIP-seq experiment allowed the identification of numerous new potential sRNAs that could be involved in the observed differences in Hfq interactomes between strains.

**The accessory Hfq interactome comprises T3SS mRNAs and a cluster of CRISPR guide RNAs.** The Type III Secretion System is the major bacterial virulence determinant, consisting of a nano-machine facilitating the injection of a small group of toxins inside of host cells[49]. The T3SS is present in PAO1- and PA14-lineages and its expression is tightly regulated at both the transcriptional and post-transcriptional levels[50]. The RIP-seq enrichment analysis resulted in the identification of nearly all T3SS genes being bound to Hfq throughout all growth phases in both strains (Fig. 5a–c), many of them being among the most enriched targets (Supplementary Data 1). The *exsA* mRNA, encoding for a major transcriptional activator of T3SS gene, was strongly enriched (Fig. 5c), in agreement with our previous findings on the role of Hfq in base pairing between sRNA sr0161 and *exsA* mRNA[13]. The T3SS genetic determinants comprise more than 30 genes encoding the T3SS machinery and its regulatory proteins and at least seven transcriptional units resulting in the transcription of long polycistronic mRNAs. The observation that Hfq binds to some of these mRNAs reveals the existence of a novel direct regulatory mechanism potentially globally affecting T3SS-related translation. Moreover, all secreted toxins utilizing T3SS, that are encoded at different genetic loci, were found among the most enriched RNAs in the Hfq RIP-seq experiment in both strains, including the shared ExoT and ExoY, as well as strain-specific toxins, ExoS (in PAO1) and ExoU (in PA14) (Fig. 5d). Therefore,

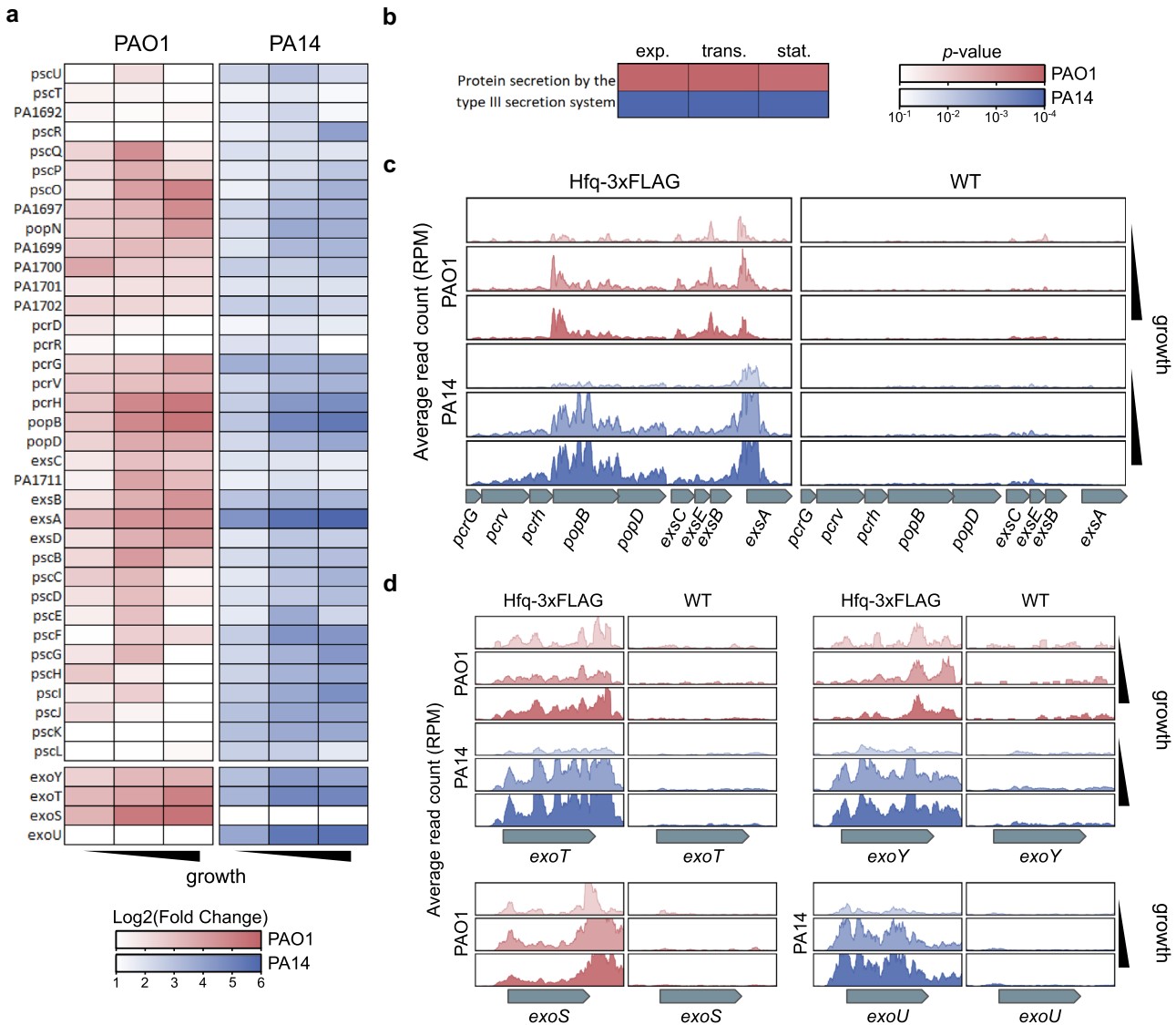

**Fig. 5 Hfq binds to most of the T3SS-related RNAs. a** Heatmap of fold enrichments in Hfq-3xFLAG samples compared to WT for all T3SS-related genes in PAO1 and PA14 in all growth phases. **b** Heatmap showing results of functional enrichment analysis for the T3SS-related GO Term for PAO1 and PA14 in all growth phases using DAVID analysis[85]. Only genes shared with PAO1 homologs were used. **c, d** Read coverages, averaged between biological replicates (*n* = 2), at the *pcrG-to-exsA* (**c**) and all exotoxins (**d**) loci for Hfq-tagged and control samples in both strains in all growth phases. Read counts are expressed in read per million reads (RPM). Source data are provided as a Source Data file.

in addition to the known Hfq influence on T3SS expression through *exsA* and *fis* mRNAs or RsmY[10,13,51], Hfq governs multiple interactions with most T3SS mRNAs, the precise regulatory role of which still remains to be investigated.

Clustered Regularly Interspaced Short Palindromic Repeats (CRISPR) systems are widely used prokaryotic antiviral defense mechanisms. Bacteria acquire resistance to bacteriophages by chromosomal integration of short fragments of viral nucleic acids serving later as RNA templates (crRNA) to interfere with phage proliferation[52]. Several CRISPR systems are present in different *P. aeruginosa* strains[53] suggesting that they were acquired through horizontal gene transfer. While the PAO1 strain does not encode any, PA14 and IHM87 each carries a CRISPR type I-F locus (*PA14_33300-PA14_33350* and *IHMA87_02728-IHMA87_02733*, respectively). The PA14 CRISPR system has been shown to be regulated by quorum sensing[54] and by the PhrS sRNA[55]. PhrS represses the Rho-dependent termination of crRNAs in the absence of Hfq, promoting CRISPR adaptive immunity against

bacteriophage infection and represents the first example of sRNA-mediated regulation of CRISPR systems. In bacterial genomes, the crRNAs clusters are usually found linked to the loci encoding CRISPR functional proteins, and PA14 and IHMA87 possess 2 and 3 crRNAs clusters, respectively. Here, the Hfq RIP-seq experiment revealed one cluster of crRNAs enriched up to 12-fold in late growth phases, specifically in IHMA87 (Fig. 6a). This cluster encodes crRNAs predicted to target various phages and unknown sequences (Supplementary Data 8). Interestingly, neither of the two other crRNA clusters in IHMA87, nor any PA14 crRNA clusters were found bound to Hfq (Fig. 6b). PA14 crRNAs were not enriched in our RIP-seq experiment, confirming the Hfq-independent nature of PhrS CRISPR regulation in this strain[55]. Additionally, we found that one spacer in the IHMA87 cluster, and none in PA14, matched the known 5 AAN repeats Hfq motif[14], which could explain the observed strain-specific binding of Hfq. This result highlights a novel major target of Hfq, which might thus play a role in the regulation of CRISPR

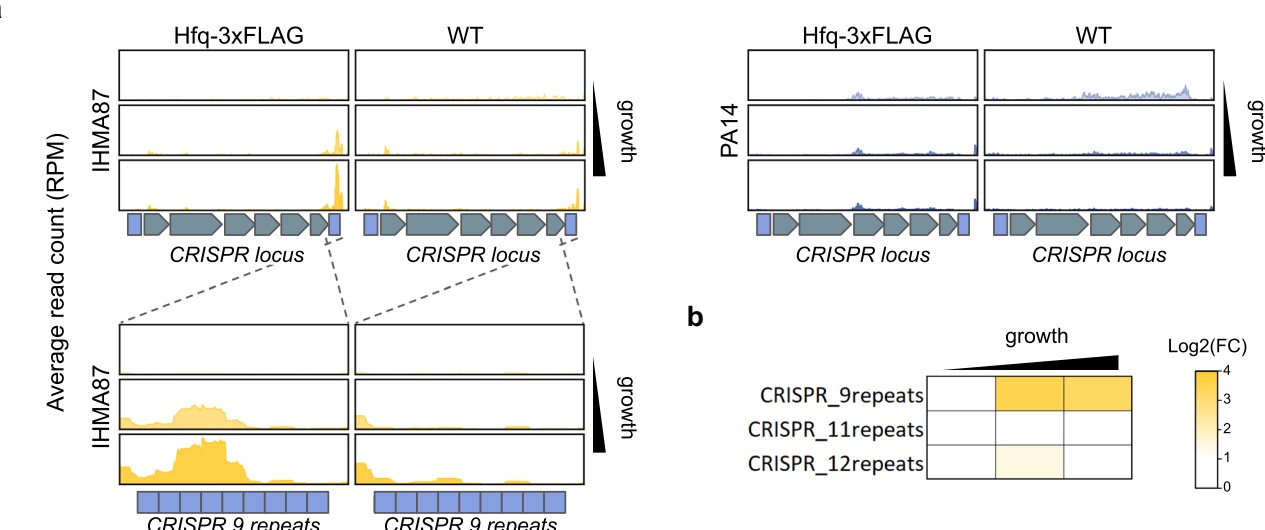

**Fig. 6 Hfq binds to a cluster of CRISPR RNA specifically in IHMA87. a** Read coverages, averaged between biological replicates ($n = 2$), at the major CRISPR loci for Hfq-tagged and control samples in IHMA87 and PA14 in all growth phases. **b** Heatmap of fold enrichments in Hfq-3xFLAG samples compared to WT for the annotated CRISPR clusters in IHMA87 in all growth phases. Read counts are expressed in read per million reads (RPM). Source data are provided as a Source Data file.

guide RNAs in a strain-specific way. Altogether our results revealed the accessory interactomes of Hfq, which comprise numerous mRNAs encoding major cellular components.

**Intraspecies differences in Hfq interactomes have various origins.** The comparison of Hfq RIP-seq results across the three PAO1, PA14, and IHMA87 strains allowed the delineation of the accessory Hfq interactome. Interestingly, while about 30% of differences in Hfq interactomes between strains were explained by differences in accessory genomes (Fig. 3d), many conserved RNAs were found to interact with Hfq specifically in one or two strains only. Two main mechanisms could explain these differences: (i) differences in RNA abundance between strains or (ii) differences in Hfq affinity to the RNA. Since it was previously proposed that RNA abundance can impact Hfq-RNA binding[14], we further explored this hypothesis. To test it, we analyzed RNA abundance using read counts from RIP-seq control samples. Indeed, as RIP-seq usually identifies full-length transcripts[56], and no specific enrichment is seen in the control samples, the read counts reflect RNA abundance in the exact same samples. The comparison between the tested strains and growth phases revealed an overall good correlation of RNA abundance between strains, and between growth phases (Fig. 7a). For instance, the overall Hfq interactome shows the highest number of differences between PAO1 and IHMA87 in the stationary phase; however the majority of RNAs exhibit similar expression patterns between these two (Fig. 7b-c). To comprehensively assess the impact of RNA abundance on the measured Hfq interactomes, we compared the cumulative distributions of abundance fold changes for differentially enriched genes and all conserved genes for all 18 strains combinations across growth phases (Fig. 7d). This analysis revealed that in the majority of cases there was no significant differences between the two groups, further supporting the idea that many differences observed between strains are not due to differences in RNA abundances. Concomitantly, this analysis also revealed which strain-specific targets are indeed due to differential RNA abundance, as it is the case for the IHMA87-specific target *pvdA*, encoding a pyoverdine biosynthetic enzyme[57], which was found significantly more abundant in IHMA87 (Supplementary Data 1 and Supplementary Fig. 5). This confirms the previously reported

higher expression of the *pvd* cluster in IHMA87 in comparison to PAO1[26], and probably explains the observed difference in RIP-seq enrichment. Overall, RIP-seq control data can be used to infer RNA expression patterns and define the origin of binding differences observed across strains or conditions, which revealed that many of the strain-specific targets identified here are not due to difference in RNA expression but probably to differences in affinity to Hfq or unknown context-specific binding regulatory mechanisms. Further work would now be required to investigate these mechanisms, such as assessing the exact location and conservation of Hfq-binding sites or the conservation of potential protein or sRNA partners.

**Conserved post-transcriptional regulation of *vfr* by quorum-sensing sRNA, RhlS.** One of the transcripts bound by Hfq and identified by RIP-seq codes for the global regulator Vfr. Hfq appears to bind the 5' leader sequence and translation initiation region (TIR) of *vfr* mRNA in all three strains and growth phases (Fig. 3f), suggesting that Hfq might be a critical factor in regulating this transcription factor. The comparison of *vfr* nucleotide sequences in the three lineages of strains showed that the two predicted Hfq-binding sites[42] are conserved in each strain (Supplementary Fig. 6). While the *vfr* sequences are identical in PAO1 and PA14, the 5' leader sequence of the same gene in IHMA87 differs from the other two at five different locations. The interaction of Hfq with the *vfr* mRNA could involve any of Hfq known mechanisms of action[4], including sRNA-dependent post-transcriptional regulation. To investigate whether this interaction involves sRNAs and if the potential sRNA regulators of *vfr* are conserved among the three strains, we carried out a reverse GRIL-seq (rGRIL-seq) analysis[23]. This method allows the identification of unknown sRNA regulators of a specific mRNA target in living cells, by ligation of the ends of sRNAs base paired with their target, catalyzed by T4 RNA ligase, and recovering of the chimeras using oligonucleotides complementary to the target sequence. After overexpression of the T4 RNA ligase in living cells during exponential or stationary phase growth, total RNAs were isolated and enriched for *vfr*-containing chimeric transcripts prior to sequencing. We obtained on average between 1327 and 5651 of *vfr*-chimeric reads from PA14, PAO1, and IHMA78

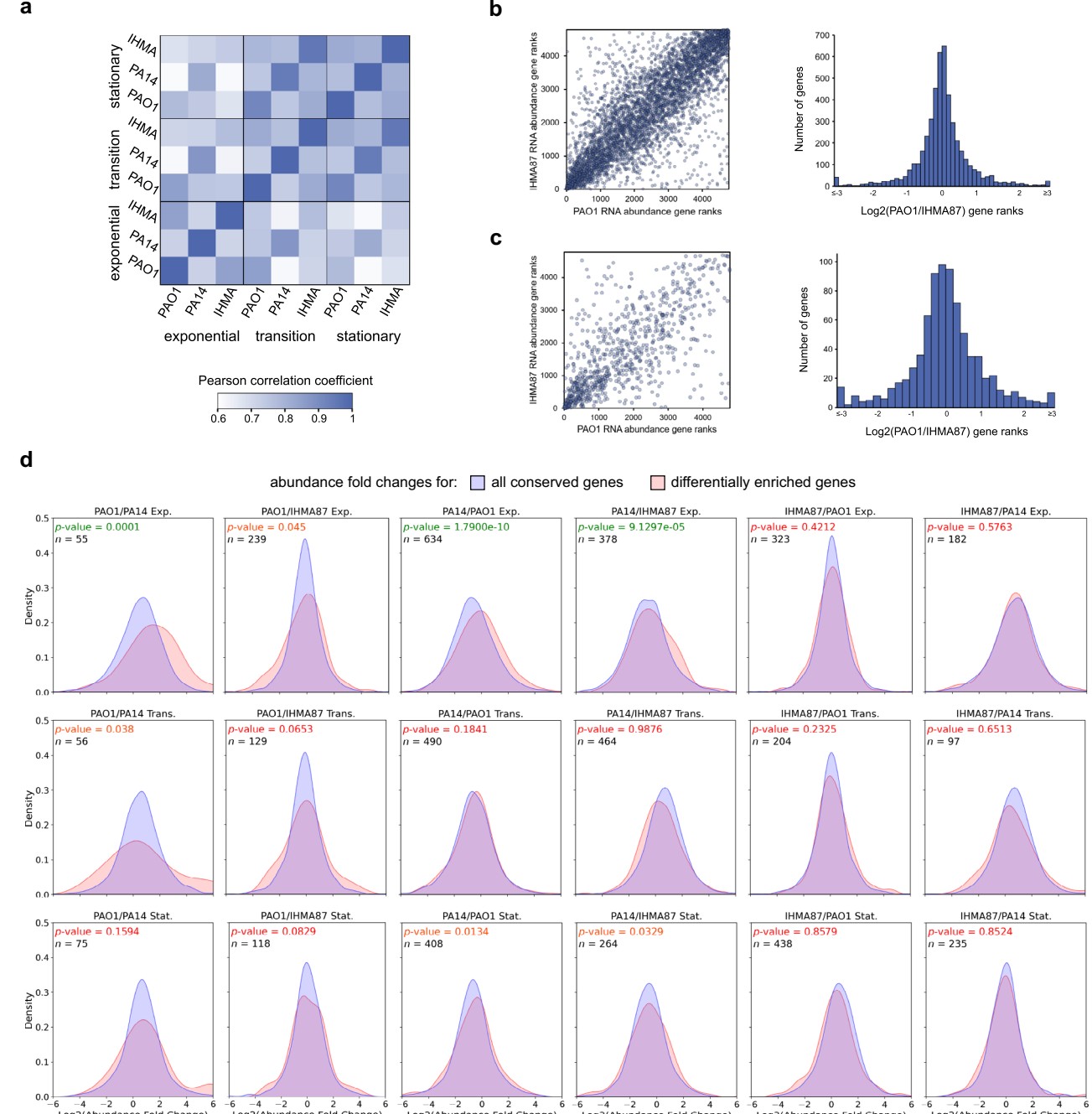

**Fig. 7 The effect of RNA abundance on interstrain enrichment differences. a** Heatmap of Pearson correlation coefficient from RNA abundance rank-matching analysis between all strains and growth phases. **b** Comparison of RNA abundance ranks for all shared genes between IHMA87 and PAO1 in the stationary phase. **c** Comparison of RNA abundance ranks for only genes differently enriched between IHMA87 and PAO1 in the stationary phase. **d** Density plots of abundance fold change distributions for all 18 strain combinations in each growth phases. Kolmogorov-Smirnov tests were used to assess if abundance fold changes were significantly higher (one-sided) for differently enriched genes (red) against all conserved genes (blue). *p*-values are reported on each corresponding plot (green: <0.01, orange: <0.05, red: >0.05) along with the number of differently enriched genes (*n*). Source data are provided as a Source Data file.

(Supplementary Data 9). Strikingly, the rGRIL-seq analyses revealed a single sRNA ligated to *vfr* mRNA in both exponential and stationary phases and in all three strains (Fig. 8a–d; Supplementary Data 10). This sRNA has been previously identified and is referred to as RhlS[58]. RhlS is co-transcribed with the *rhlI* gene, encoding a *N*-Acyl homoserine lactone (AHL) synthetase involved in quorum sensing, and terminated as a 70-nt RNA located at the 5' leader sequence of *rhlI* gene. This sRNA was found to be required for the production of normal levels of AHL

through activation of *rhlI* translation and is thus important for quorum sensing[58]. The sequence of this sRNA is identical between PAO1 and PA14, while the same transcript in IHMA87 differs by substitution of three adjacent nucleotides and a single nucleotide deletion at the end of the terminator sequence (Supplementary Fig. 6b). The base-pairing prediction between RhlS and *vfr* mRNA using IntaRNA[59] showed a 11-consecutive base-paired nucleotides between the two (Fig. 9a). The predicted base-paring sequences of the three *vfr* transcripts are located in the

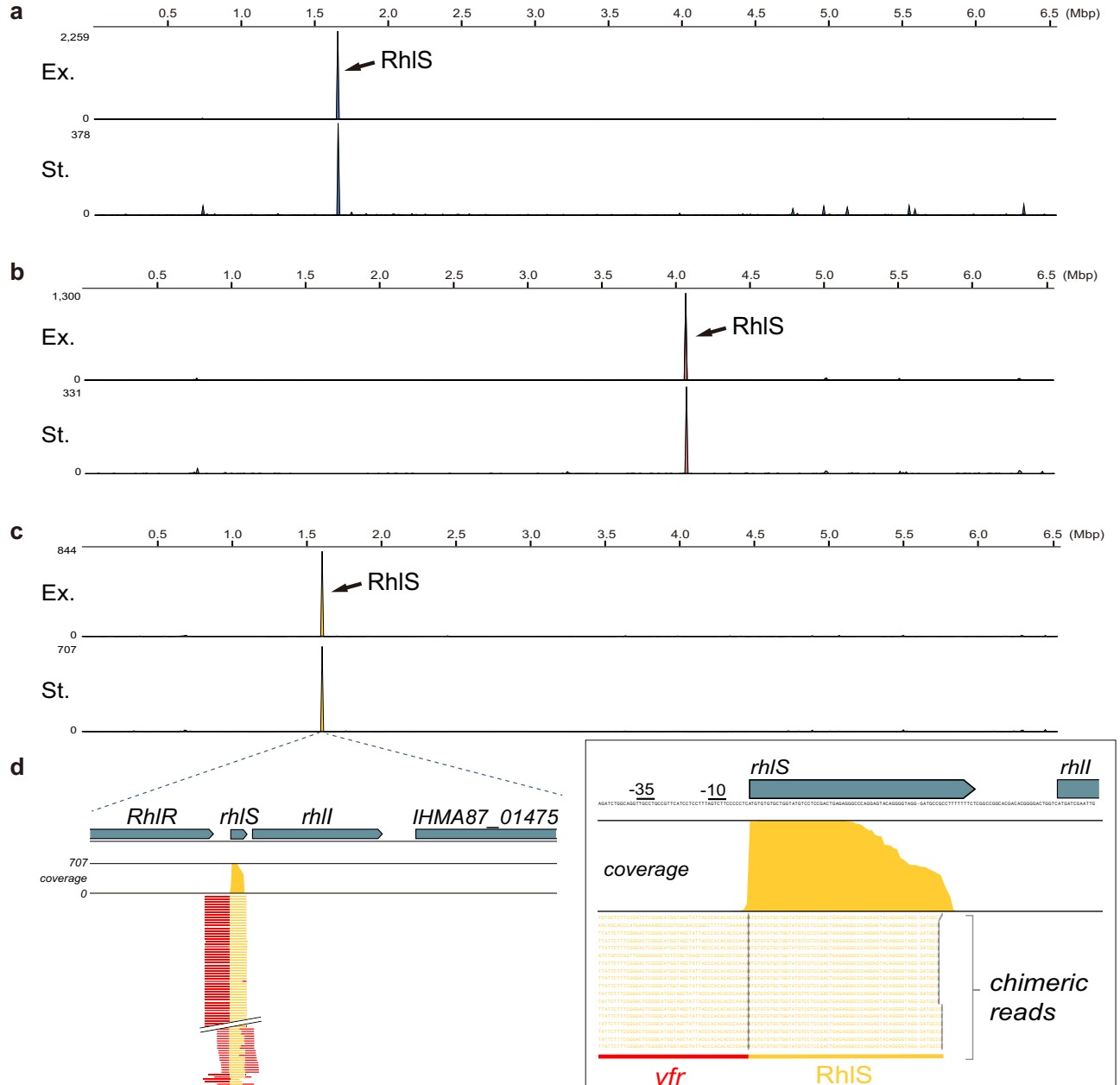

**Fig. 8 Identification of RhlS interacting with the *vfr* mRNA by rGRIL-seq.** The *vfr*-chimeric RNA reads are mapped on the respective chromosomes in *P. aeruginosa* strains PA14 (**a**), PAO1 (**b**), and IHMA87 (**c**) during exponential (Ex.) or stationary (St.) phase growth. Arrows point the significant accumulation of *vfr*-chimeric RNAs at *rhlS* locus in actively dividing cells (OD$_{600}$ of ~0.4), refer to it as exponential (Ex.) and high density (OD$_{600}$ of ~3.5), refer to it as a stationary (St.) phase cultures. One of the rGRIL-seq results in replicate experiments was represented for each strain. **d** Enlarged images of the peak formed by mapping *vfr* chimeras at *rhlS* locus in IHMA87.

early 5' leader sequence (−139 to −129), while those in RhlS are at 5' region of the sRNA (+5 to +15), which are also conserved in all three strains (Supplementary Fig. 6b).

To confirm the role of RhlS in *vfr* regulation, we used two approaches. First, we constructed an IPTG-inducible translational fusion between the 5' leader sequence, the first 30 codons of Vfr and *lacZ*; this construct was integrated into the chromosome of PA14Δ*rhlS* (Fig. 9b). Since RhlS is known to negatively regulate the expression of *fpvA*[58], encoding a siderophore receptor, we used a translational *lacZ* fusion to this gene as a control (Fig. 9c). After introducing arabinose inducible RhlS expression vectors in these strains, we monitored the activity of β-galactosidase at three time points (2, 5, and 7.5 h) after induction with IPTG and arabinose

(Fig. 9c). While the β-galactosidase activity was reduced by more than 50% in the strain carrying *fpvA*::*lacZ* fusion as expected, it was also increased in the strain carrying *vfr*::*lacZ* fusion (Fig. 8c), suggesting an activating role of RhlS on *vfr* translation. We then further examined this interaction by introducing 4-nucleotide mutations in the sRNA (RhlS (M)) and the complementary base pair changes in the *vfr* mRNA (*vfr* (m)) at the predicted interaction sites (Fig. 9a). Whereas the mutations in RhlS (M; CUGG to GACC) fully abolished the observed regulation, it was restored by the introduction of the compensatory mutation in *vfr* (m; CUGG to GGUC), confirming the specific interaction (Fig. 9d). Next, to further examine the effect of RhlS on Vfr protein levels, we performed western blots using antibody raised against Vfr, sampling at several

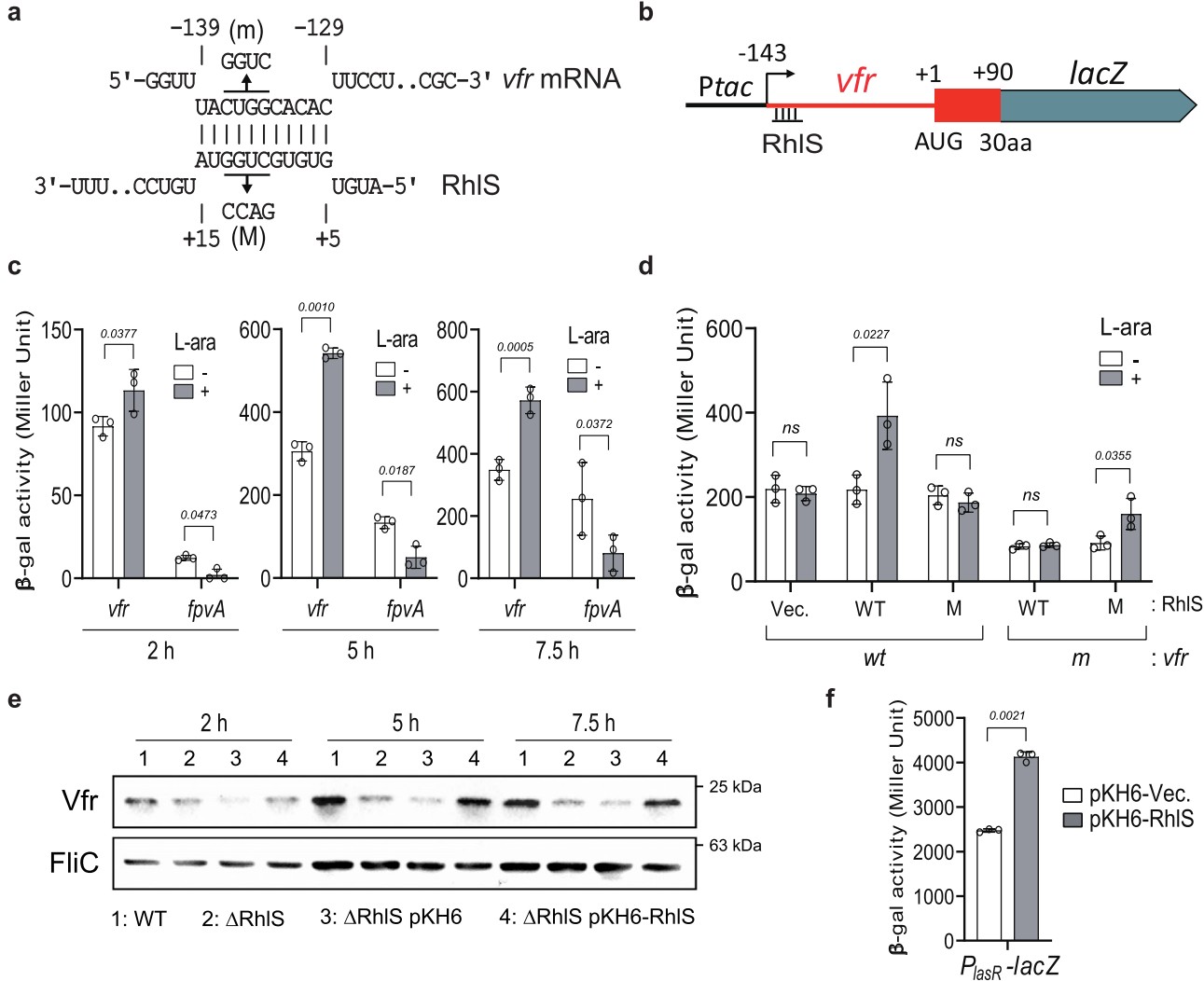

**Fig. 9 RhlS activates *vfr* at post-transcriptional level. a** Base-pairing model of RhlS interaction with *vfr* mRNA (Hyb. Energy; -16.96 kcal/mol) generated by IntaRNA[59]. **b** *vfr::lacZ* fusion construct used for β-galactosidase reporter assays. The sequences (red) containing the 5' leader sequence and the codons for the first 30 amino acid of Vfr were fused with *lacZ* gene to create *vfr::lacZ* translational fusion using P*tac* inducible pminictx-*lacZ*[60]. The constructs were introduced into PA14 Δ*rhlS* followed by plasmid (pKH6-rhlS). The approximate location of the predicted base pairing site of RhlS on *vfr* mRNA is shown at the 5' leader sequence of the *vfr::lacZ* fusion. **c** Relative β-galactosidase activities under RhlS non-inducible condition (white bars) and inducible condition (gray bars) are shown at three post-induction time points (2 h, 5 h, and 7.5 h) in PA14 Δ*rhlS* strains carrying the *vfr::lacZ*, *fpvA::lacZ* and the RhlS expression vector. The activity was measured at three time points after sRNA induction at $OD_{600}$ = 0.1–0.2. L-arabinose and IPTG were added to express the sRNA and the *lacZ* fusion, respectively. Data are shown as mean ± SD for three biological replicates. Statistical comparisons were performed using two-sided Student's *t*-test. Exact *p*-values are shown. **d** Relative β-galactosidase activities in response to wild-type (WT), mutant (M) RhlS, or the compensatory mutant in the 5' leader sequence of *vfr* (m). Data are shown as mean ± SD for three biological replicates. Statistical comparisons were performed using two-sided Student's *t*-test. Exact *p*-values are shown; ns not significant. **e** Immunodetection of Vfr in bacterial cell extracts. Total proteins were sampled at three time points following induction of expression of RhlS with 0.2% L-arabinose followed by SDS-PAGE, Western blotting and probing with antibody to Vfr or FliC (loading control). Results are from one representative experiment out of three. **f** β-galactosidase activity from the $P_{lasR}$-*lacZ* transcriptional fusion in *rhlS* mutant strain containing empty vector (pKH6) or RhlS expression (pKH6-RhlS). Data are shown as mean ± SD for three biological replicates. Statistical comparisons were performed using two-sided Student's *t*-test. Exact *p*-values are shown. Source data are provided as a Source Data file.

time points during growth. Compared to the wild-type *P. aeruginosa*, the amount of Vfr was significantly reduced in the *rhlS* mutant, particularly during transition and stationary phases (Fig. 9e) and was restored upon complementation of RhlS from an L-arabinose inducible RhlS expression vector. As a downstream effect of the RhlS-dependent regulation of *vfr*, we examined the transcriptional activity of *lasR* since it is known to be directly regulated by Vfr in a cAMP-independent manner. Using a *lasR* chromosomal transcriptional reporter, we showed that *lasR* promoter activity was activated by the expression of RhlS (Fig. 9f), suggesting that RhlS is sufficient to activate *lasR* through Vfr. Altogether, these results

indicate that RhlS is a positive regulator of *vfr* expression by directly interacting with its 5' leader sequence, which impacts Vfr function as a global regulator and its downstream regulatory effects.

## Discussion

RBP-driven post-transcriptional regulatory networks in the major human pathogen *P. aeruginosa* have been the topic of numerous studies over the last decade[9,13,14,60,61], in all cases on one single strain. However, strains of *P. aeruginosa* can be grouped into three major phylogenetic lineages[15], each displaying different

genotypic and phenotypic features. In this study, we provide a first glance at the intraspecies post-transcriptional regulatory diversity by comparative RIP-seq analysis using the RNA chaperone Hfq. Our approach showed that most Hfq interactions are not conserved between three strains of *P. aeruginosa*. Hfq interacts with 13 to 21% of all annotated RNAs, spanning conserved major biological processes and hundreds of strain-specific targets.

In this context, the use of RIP-seq allowed the global investigation of Hfq interactomes. While this technique greatly facilitates such analysis by the enrichment of full-length RNAs, preventing the need of prior knowledge on untranslated regions, it also has limitations. Indeed, this technical feature makes the precise identification of the binding site location impossible, unlike with CLIP-seq where the bound RNAs are purposely shortened around the bound protein[31]. In the light of the large differences in Hfq interactomes that are not explained by different genome contents or RNA abundances between strains, future works should focus on investigating potential sequence differences at the binding sites.

Hfq regulates bacterial virulence through multiple direct and indirect mechanisms[10,11,36]. Our analysis identified both known and novel virulence-related regulatory targets for Hfq in the different strains. First, the *vfr* mRNA, encoding for a global virulence regulator[62], was strongly bound to Hfq in all three strains. Interestingly, while Vfr is known to regulate the T3SS[63], it has been recently shown to directly regulate the two-partner secretion system ExlBA[64], which is the main virulence factor in the T3SS-lacking PA7/IHMA87-like lineage[19,20]. We applied rGRIL-seq, a method for identifying regulatory sRNAs of specific mRNAs, to all three strains and have shown that they share a common sRNA (RhlS) regulator of *vfr* expression. RhlS has been shown to be linked to the expression of homoserine lactone synthetase RhlI and is itself regulated by quorum sensing[58], its interaction with *vfr* mRNA connects quorum sensing to virulence regulation through a post-transcriptional mechanism. Based on our predicted base-pairing model and complementation analysis, we infer that RhlS is base pairing with the complementary sequence of *vfr* 5' leader sequence and this interaction seems to be required for activation of *vfr* translation. As shown in Fig. 9d, the mutation of RhlS abolishes the activation while the compensatory mutation in the interaction site of *vfr* mRNA restores the activation. Several mechanisms could explain these observations. The 5' leader sequence of *vfr* mRNA was proposed to form a stem-loop structure encompassing the Hfq-binding sites, RsmA-binding sites and ribosomal binding sites. RhlS could facilitate the binding of Hfq on the the stem-loop by rearranging the mRNA structure, as shown for several positive sRNA regulators (DsrA for *rpoS* regulation[65] and RyhB for *shiA* regulation[66]). Alternatively, RhlS could protect the *vfr* mRNA against RNases (exo- or endo-nuclease), as shown for RydC, which protects *cfa1* mRNA against RNase E[67]. This reveals an interesting feature of regulatory plasticity, as the conserved interactions between Hfq, *vfr* mRNA and RhlS result in different virulence-related regulatory outputs (T3SS or ExlBA) due to the diversity found in Vfr targets across *P. aeruginosa* lineages. Additionally, the T3SS is directly regulated by Hfq through stabilization of the interaction between *exsA* mRNA, encoding the T3SS transcription regulator, and the sr0161 sRNA[13]. Here, we show that in addition, Hfq interacts with most of the T3SS-related mRNAs, including those of all four T3SS toxins. The role of Hfq in these interactions remains to be explored. Indeed, while the most studied function of Hfq is to facilitate regulatory mRNA-sRNA interactions, it can also play other sRNA-independent roles, including modulation of mRNA stability and translation[8]. Altogether, our study reveals several new regulatory pathways through which Hfq regulates directly or indirectly the two major virulence factors, the T3SS

and ExlBA, as well as many other virulence-related genes through Vfr regulation.

Our study also revealed a cluster of CRISPR RNAs as Hfq accessory targets, specifically in the IHMA87 strain. Although the leader sequence of CRISPR2 crRNAs were found to interact with a sRNA in PA14[55], our study showed that this interaction was Hfq-independent, suggesting diverse mechanisms governing crRNA transcription and stability. The fact that the enrichment signal spans several crRNAs in the IHMA87 cluster suggests a binding of Hfq to the unprocessed guide RNA. In light of the importance that CRIPSR systems have taken in many biotechnology fields over the last few years[68], these results call for further investigation of the underlying mechanism of crRNAs stabilization revealed here.

In conclusion, we showed that the RBP-driven post-transcriptional regulatory networks exhibit intraspecies diversity, which could potentially explain phenotypic differences. We identified numerous specific and common interactions between the three lineages and propose several possible underlying mechanisms, including (i) differences in gene or sRNA content or expression, (ii) differences in affinities to Hfq, and (iii) strain-specific role of the target, as illustrated for Vfr which has different regulons between strains. All three mechanisms were found responsible for major differences in Hfq role across *P. aeruginosa* lineages and should be investigated in more details in the context of regulatory networks plasticity. Altogether, this work highlights the importance of considering regulatory and genomic content of specific strains when studying regulatory networks, and calls for more comparative approaches in the fields of transcriptional and post-transcriptional regulation.

## Methods

**Bacterial strains**. The list of bacterial strains created for this study is provided in Supplementary Data 11. For engineering chromosomal deletions of genes in *P. aeruginosa*, the pEXG2 plasmid was used, as described previously[69]. The pmi-niCTX-Ptac-t4rnl1 plasmid was used for chromosomal integration of the T4 RNA ligase gene at the φCTX site of *P. aeruginosa*. Plasmids were introduced into *P. aeruginosa* by conjugation using *E. coli* SM10λpir or by triparental mating using *E. coli* DH5a (pRK2013) as a helper plasmid[70]. *P. aeruginosa* and *E. coli* strains were grown in Lysogeny Broth (LB) at 37 °C under agitation or on plates, in medium consisting of LB with 1.5% agar. When appropriate, the media was supplemented with antibiotics: gentamicin at 75 µg/ml (25 µg/ml for *E. coli*) and irgasan at 25 µg/ml for *P. aeruginosa*, tetracycline at 75 µg/ml for *P. aeruginosa* and 10 µg/ml for *E. coli*.

**Plasmids and genetic manipulations**. The plasmids and oligonucleotides used in this study are listed in Supplementary Datas 11 and 12, respectively.

For Hfq C-terminal tagging, ~500 bp of upstream and downstream sequences flanking *hfq* stop codon were amplified from each of three genomes, using primers hfq-3xFLAG-sF1, hfq-3xFLAG-sR1, hfq-3xFLAG-sF2, hfq-3xFLAG-sR2, and hfq-3xFLAG-IHMA-sR2, carrying the sequence coding for a 3xFLAG tag (DYKDHD GDYKDHDIDYKDDDDK) and a 3-Glycine linker. The two resulting, overlapping fragments were then cloned into *Sma*I-cut pEXG2 by SLIC[71]. After cloning, all plasmids were transformed into competent TOP10 *E. coli* cells and verified by sequencing. The pEXG2-derived vectors were transferred into *P. aeruginosa* strains by triparental mating using pRK600 or pRK2013 as helper plasmids. Merodiploids resulting from cointegration events were selected on LB plates containing irgasan and gentamicin. Single colonies were then plated on NaCl-free LB agar plates containing 10% (wt/vol) sucrose to select for the loss of plasmid, and the resulting sucrose-resistant strains were checked for mutant genotype and gentamicin sensitivity.

For chromosomal expression of T4 RNA ligase, the pminiCTX-Ptac-t4rnl1 plasmid was constructed by inserting the T4 RNA ligase gene (*t4rnl1*) into an IPTG-inducible vector, pminiCTX-Ptac (Tet[R]). The T4 RNA ligase gene was amplified from pET16β-*t4rnl1* (generously provided by Ushati Das, Stewart Shuman's lab) using primer pair F_ERI_MMB_T4RL and R_BHI_MMB_T4RL, and then ligated with the linearized plasmid, pminiCTX-Ptac (*Eco*RI/*Bam*HI-digested) using an In-Fusion Cloning kit (Clontech, cat# 639648). The pminiCTX-Ptac plamid was constructed by inserting a DNA segment containing the *lacIq* gene, the IPTG-inducible promoter, multicloning sites and terminator sequence into the linearized pminiCTX-1 vector[72] (*Kpn*I/*Sac*I-digested, Tet[R]). This segment was an amplicon generated by PCR using the primer pair F_KpnI_pMMBIq and R_sacI_pMMBTer and plasmid pMMB67EH as a template[73]. To construct the vector for expressing the

*P. aeruginosa* RhlS sRNA, its gene was amplified by PCR with the primer pair F_xbaI_pkh6_RhlS+1 and R_Hd3_pkh6_RhlS+100 and the *P. aeruginosa* PA14 chromosomal DNA as the template and inserted into pKH6 (Gm$^R$) following digestion with XbaI and HinDIII. For the RhlS mutant (RhlS (M)), the same cloning procedure was carried out as described in cloning for wild-type RhlS except for using a different forward primer (F_xbaI_pkh6_sRhlI_M) at the PCR amplification step. For chromosomal expression of *vfr::lacZ* or *fpvA::lacZ*, the sequences from the transcription start site to the 30th amino acid codon of each gene were amplified by PCR with the primer pairs F_ERI_vfr-Z30aa and R_Hnd3_vfr-Z30aa or F_ERI_fpvA-Z30aa and R_Hnd3_fpvA-Z30aa and the *P. aeruginosa* PA14 chromosomal DNA as template. The DNA segment was inserted in the EcoRI/HinDIII-digested pminiCTX-P*tac::lacZ*[60]. For compensatory *vfr* mutant reporter (*vfr* (m)), the same cloning procedure was carried out as described in the cloning for wild-type *vfr* except for using a different forward primer (F_ERI_vfr-Z30aa_m) at the PCR amplification step. For the *lasR-lacZ* transcriptional fusion, a *lasR* promoter fragment including Vfr binding site was amplified by PCR from PA14 genomic DNA using primers F_ F_lasR_ERICTXlacZ-264 and R_lasR_BHICTXlacZ+236. The DNA fragment was cloned into the integration vector mini-CTX-lacZ[72] and the vector integrated into the chromosome of PA14 Δ*rhlS* strain. Stellar E. coli cells (Clontech, cat# 636763) were used as competent cells in the cloning procedure.

**Homolog genes/sRNAs identification and phylogeny**. All *P. aeruginosa* complete genomes (*n* = 192) were retrieved from the *Pseudomonas* Genome Database[30]. Homolog identification was performed by Reciprocal Best Blast Hit (RBBH) analysis[74] on the European Galaxy server[75] using all protein sequences from *P. aeruginosa* PA14 against all protein sequences from the other 191 genomes with minimum percentage alignment coverages of 90 and sequence identity of 50. The sequences from 66 core genes were concatenated for each genome and a multiple alignment was performed with MAFFT Galaxy version 7.221.3 using default settings[76], as previously described[77]. The resulting alignment was used to build a Maximum-Likelihood phylogenetic tree using MEGA X[78] with 100 bootstraps, which was visualized and annotated using iTOL v5[79]. Annotations for 200 PA14 sRNAs were obtained from Wurtzel et al., 2012 and used to retrieve the corresponding sRNAs sequences. sRNA homologs were identified in *P. aeruginosa* complete genomes using megablast (blastn Galaxy version 0.3.3) with minimum percentage alignment coverages of 90 and sequence identity of 80. All newly identified sRNAs were added to the PAO1 and IHMA87 genome annotations that were used in this work. Hierarchical clustering analysis of the 200 sRNAs conservation in *P. aeruginosa* genomes was performed with BioVinci v1.1.5 (BioTuring Inc.) using Ward's minimum variance and Manhattan distances.

**Western immunoblotting**. For the detection of Hfq-3xFLAG, cells were grown at 37 °C until an OD$_{600}$ of 1.0 after dilution from overnight precultures. For the detection of Vfr and FliC, the overnight culture of *P. aeruginosa* PA14 (wild-type) or PA14 Δ*rhlS* carrying pKH6-RhlS in LB at 37 °C were diluted to an optical density measured at 600 nm (OD$_{600}$) of ~0.01 in fresh medium and further grown until OD$_{600}$ of ~1.0. L-arabinose was then added to a final concentration of 0.2% (w/v) and cultures were further incubated at 37 °C for 2, 5 or 7.5 h. Then, 300 μl from each culture was centrifuged and the pellets were resuspended in PBS to reach 5 OD$_{600}$. After addition of 4× loading buffer (Bio-Rad, cat# 1610747), samples were incubated at 98 °C for 15 min. Samples were then loaded on an SDS-PAGE gel (Criterion TGX, 4–20%, Bio-Rad, cat# 5678093). The gels were blotted onto nylon membranes (Bio-Rad, cat# 1704159) using Trans-Blot® Turbo™ Transfer System (Bio-Rad, cat# 1704150) and probed using primary anti-FLAG M2 mouse monoclonal antibodies (1:2000 dilution) (Sigma, cat# F1804), anti-Vfr (1:25,000 dilution) and anti-FliC (1:5000 dilution) rabbit antibodies[64]. Secondary antibodies were anti-mouse-HRP (1:80,000 dilution) (Sigma, cat# A9044) or anti-rabbit-HRP (1:50,000 dilution) (Sigma, cat# A0545). Membranes were developed with Luminata Classico Western HRP (Milipore, cat# WBLUC0500).

**RT-qPCR**. RNA was extracted as described for RIP-seq and treated with the TURBO DNA-free kit following manufacturer's instructions (Thermo Fisher, cat# AM1907). For RT-qPCR, 4 μl of DNase-treated RNA were used with the Luna Universal One-Step RT-qPCR kit (NEB, cat# E3005) and primers specific to PrrF1 or 6S RNA (Supplementary Data 12). Experiments were performed in biological duplicates for each strain, and the ΔΔCq method was used to obtain fold change values between WT and Hfq-tagged strains normalized to the 6S RNA abundance.

**Hfq-RNA co-Immunoprecipitation**. Bacterial cultures were started in 200 ml of LB medium at OD$_{600}$ = 0.1 from overnight cultures. At time of sampling (after 200, 400, or 600 min of incubation), the equivalent of 50 ml of OD$_{600}$ = 1 was pelleted by centrifugation and snap-frozen in liquid nitrogen. Pellets were then resuspended in 700 μl of Lysis buffer (20 mM Tris-HCl pH 8, 150 mM KCl, 1 mM MgCl$_2$, 1 mM DTT) and, after addition of 700 μl of glass beads (Sigma, cat# G4649), vortexed by cycles of 30 seconds vortexing and 30 seconds on ice for a total of 10 min. Lysates were centrifugated for 30 min at 16,000 × *g* at 4 °C and supernatant were then incubated with 40 μl of washed Anti-FLAG M2 Magnetic beads (Sigma, cat# M8823) for 1.5 h at 4 °C. The beads were then washed five times by resuspension in

500 μl of Lysis buffer. After the last wash, 500 μl (1:1 volume) of 125:24:1 acid-phenol:chloroform:isoamyl alcohol pH 4.5 (Thermo Fisher, cat# AM9720) was added to the resuspended beads and vortexed for 10 s. After 2 min of incubation at room temperature, samples were centrifuged for 30 min at 16,000 × *g* at 4 °C and the upper phase was recovered and used for overnight RNA precipitation at −20 °C in 1.5 ml (3:1 volume) of ice-cold pure ethanol containing 50 μl of 3 M sodium acetate (Thermo Fisher, cat# AM9740) and 1 μl of GlycoBlue coprecipitant (Thermo Fisher, cat# AM9515). Pelleted RNA was then washed with 70% ethanol and resuspended in 20 μl of TURBO DNase mix, incubated following manufacturer's instructions (Thermo Fisher, cat# AM2238) and further cleaned up by an additional phenol:chloroform extraction followed by overnight ethanol precipitation, as described above. RIP-seq was also conducted on WT strains in each condition as negative controls. All RIP-seq experiments were performed in biological duplicates for a total of 36 samples.

**Library construction and sequencing**. The quantity and quality of purified RNAs were then assessed on an Agilent Bioanalyzer and 1-5 ng of RNA were then used for library construction using the NEBNext Ultra II Directional Library Prep kit (NEB, cat# E7760). Sequencing was performed at the Biopolymers Facility at Harvard Medical School (https://genome.med.harvard.edu/) on an Illumina NextSeq500 for the 24 PAO1 and IHMA87 samples, and at the sequencing core facility of I2BC (http://www.i2bc.paris-saclay.fr) on an Illumina NextSeq500 for the 12 PA14 samples, with a total average of ~8.4 million reads per sample.

**RIP-seq data analysis**. Raw reads were trimmed for adapter sequences and quality using Trimmomatic[80] (Galaxy version 0.38.0) and aligned to their corresponding genome with Bowtie2[81] (Galaxy version 2.3.4.3) with default parameters. sRNAs from our genome screen (Fig. 1c) were added to the annotation files used in the analysis (Supplementary Data 3). Read counts per feature were then obtained with htseq-count[82] (Galaxy version 0.9.1). Differential RNA enrichment between Hfq-tagged and wild-type strains was assessed using DESeq2[83] (Galaxy version 2.11.40.6). Similarity between replicates was assessed through Pearson correlation (Supplementary Fig. 2). Enriched group comparisons between growth phases were done by hierarchical clustering analysis using BioVinci with Ward's minimum variance and Euclidian distances. For coverage visualization, Bowtie2 alignment output files were converted to bedgraphs with the Genome Coverage tool from BEDTools[84], counts were transformed to reads per million (RPM), and averaged between biological replicates. Average local RPM counts were used to generate coverage plots using matplotlib in Python 3. RNA abundance comparisons were done using averages of DESeq2 normalized read counts from WT samples transformed to RPM counts among the 4776 genes shared by all three strains and displayed using GraphPad Prism 7.04. Genes were ranked on their RNA abundance based on their RPM values and a rank-matching analysis was performed for all pairs of strains in all growth phases using Pearson's correlation test. Abundance fold changes between pairs of strains in each growth phase were used to identify outlier genes with significantly different RNA abundance between strains. For complementary analysis, peak calling was done using PEAKachu version 0.1.0.2 (https://github.com/tbischler/PEAKachu) with maximum insert sizes of 60. The closest gene and distance from it were identified for each peak using the ClosestBed tool from BEDTools[84]. Peaks that did not correspond to a gene found enriched with the DESeq2 method either corresponded to mRNA targets that were missed or to new unannotated sRNAs. For detection of potential new intergenic sRNAs, enriched sequences, extended by 10 bp in both directions using BEDTools SlopBed, were obtained from strictly intergenic peaks using BEDTools GetFastaBed[84]. Prediction of sRNAs was performed using StructRNAfinder[46] on the obtained sequences. Potential new antisense sRNAs were manually curated from intragenic peaks that were detected on the opposite strand of their corresponding gene found with ClosestBed.

**rGRIL-seq**. The three *P. aeruginosa* strains (PA14, PAO1, and IHMA87) carrying the IPTG-inducible T4 ligase gene (minictx-Ptac-*t4rnl1*) were grown overnight in LB containing 75 μg/mL tetracycline at 37 °C. The cultures were diluted to OD$_{600}$ = 0.01 and further grown aerobically at 37 °C. For induction of T4 RNA ligase, IPTG was added to 1 mM when cultures reached low (OD$_{600}$ of ~0.4) or high cell density (OD$_{600}$ of ~2.0). After 60 min post-induction, total RNA was isolated using the Direct-zol™ RNA MiniPrep kit (Zymo, cat# R2051), following manufacturer's instructions with some modifications. 5 OD$_{600}$ units of culture were collected by centrifugation (13,000 × *g*, 40 s), supernatants were discarded and pellets immediately snap-frozen in liquid nitrogen. Seven hundred microliters of TRI Reagent® was added to and to each frozen pellet in a 1.5 mL centrifuge tube and cells were immediately lysed using a Vortex mixer for 2 min. Cell debris were removed by centrifugation (13,000 × *g*, 1 min) and 650 μL of the lysate were transferred into a fresh tube containing the same volume of 100% ethanol. Total RNA was then purified through two Zymo-Spin™ IIC columns. Twelve micrograms of total RNA were used for enrichment of chimeric RNAs. For the enrichment of *vfr*-chimeric RNAs, four different DNA oligonucleotides were designed and 20 pmol (5 pmol each) of oligonucleotides were mixed and used in enrichment step of rGRIL-seq[23]. The enriched chimeric RNAs were precipitated at −80 °C, collected by centrifugation (10 min, 21,000 × *g*, 4 °C) and washed with 70% of ethanol (0.5 mL). The enriched RNAs were dissolved in 13 μL of nuclease-free water and

100 ng were used for the RNA library preparation using NEBNext Ultra Directional RNA Library Prep Kit (NEB, cat# E7420s). After 11 PCR cycles for amplification, libraries were then purified twice with AMPure XP beads (Beckman Coulter, cat# A63880) using a 0.9× ratio. Sequencing was performed on a Next-Seq500 instrument (Illumina) with 150 bp paired-end reads. Data analysis for rGRIL-seq was performed using CLC Genomic Workbench 7.0 as used for GRIL-seq[23,60]. The *vfr* reference sequences were composed of *vfr* CDS, 5' and 3' UTRs from each genome (PA14 (NC_008463), 716,972 to 717,799; PAO1 (NC_002516), 705,988 to 706,815; IHMA87 (NZ_CP041354.1), 672,875 to 673, 700). To create each *vfr*-deleted reference genome, the sequences including *vfr* and part of the flanking genes were deleted from each genomic reference genome (PA14, 716,451 to 718,140; PAO1, 705,706 to 707,168; IHMA87, 672,620 to 674,060).

**β-galactosidase assay.** An overnight culture of *P. aeruginosa* PA14 Δ*rhlS* (pPtac-miniCTX-*vfr*, *fpvA*, or *vfr* (m)::*lacZ*) carrying pKH6-RhlS or pKH6-RhlS (M) was grown in LB with tetracycline and gentamicin. The overnight culture was diluted to an OD$_{600}$ of ~0.02 in 1 mL of the same medium and grown to an OD$_{600}$ of ~1.0. To express the sRNA and mRNA, L-arabinose and IPTG were added to the final concentrations of 0.2% (w/v) and 25 μM, respectively. β-galactosidase activity was measured using 100 μL of cells after 2, 5 and 7.5 h induction with L-arabinose and IPTG. For the β-galactosidase activity of *lasR-lacZ* fusion carrying pKH6-RhlS, the assay was carried out by adding 0.2% (w/v) L-arabinose at an OD$_{600}$ of ~1.0 and the activity was measured 5 h post-induction.

**Interstrain comparison and functional enrichment analysis.** Results from the RBBH analysis were used to generate lists of corresponding gene IDs between the PAO1, PA14, and IHMA87 genomes. For functional enrichment analysis, only genes with PAO1 homologs could be used for PA14 and IHMA87. The resulting lists were analyzed with DAVID v6.8[85] against each corresponding strain functional background using PAO1 KEGG Pathways and GOTerm Biological Processes annotations with default parameters.

**CRISPR spacer prediction and annotation.** Clusters of CRISPR crRNAs were detected using CRISPRCasFinder with default parameters[86]. Spacer target prediction was done with CRISPRTarget with a score cutoff of 28[87].

**Reporting summary.** Further information on research design is available in the Nature Research Reporting Summary linked to this article.

## Data availability
RIP-seq and rGRIL-seq data are available under the GEO accession numbers GSE171056 and GSE171893, respectively. Genomes annotations are from the Pseudomonas database [https://www.pseudomonas.com/] (IHMA87 genome under the ID AZPAE15042). Source data are provided with this paper.

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

## Acknowledgements

We are grateful to Yanjie Chao for his technical advices on RIP-seq. Work in SL's laboratory was supported by the grant AI136789 from the NIH. The work in IA's laboratory was supported by "Fondation pour la Recherche Médicale (FRM)" [Team FRM2017, DEQ20170336705]. J.T. received a Ph.D. fellowship from the French Ministry of Education and Research and a mobility allowance from the Initiative of Excellence (IDEX), Université Grenoble Alpes for this project. The *Pseudomonas aeruginosa* strain IHMA879472 was kindly provided by International Health Management Association (IHMA; USA).

## Author contributions

J.T., I.A., and S.L. conceived and designed the experiments. J.T. and K.H. performed the experiments and analyzed the data. J.T., K.H., I.A., and S.L. wrote the manuscript.

## Competing interests

The authors declare no competing interests.

**Additional information**

