## [Peer Review File · Nature Communications]

Reviewers' Comments:

Reviewer #1:

Remarks to the Author:

Trouillon et al. examined the conservation of small, regulatory RNAs (sRNAs) and RNA binding proteins (RBPs) across *Pseudomonas aeruginosa* lineages and assayed the interactomes of the RNA binding protein Hfq in laboratory strains PAO1, PA14 and IHMA87. The authors then examined the interaction of the mRNA encoding the global regulator Vfr with the RhIS sRNA in more detail. This comprehensive study represents a significant amount of work and is an important contribution to the field, providing useful datasets for investigators interested in sRNA and RBP evolution. Most of my comments are editorial in nature.

1. Given the amount of data presented in this study, the text of the first results sections should be tightened to avoid having too many lists of numbers and to better highlight the main conclusions. I also would have liked more analysis of the factors that contribute to the strain differences in the Hfq interactomes since the differences "cannot be explained by different RNA abundances". All or parts of Supplementary Figure 6 should be included in the main text.

2. The authors should similarly be judicious about what they include in their figures.

--Figure 2: I suggest moving the heatmaps (for which the shades are difficult to distinguish) to supplementary information and substituting browser images documenting examples of changes across growth.

--Figure 3: Panels d and e are not very informative.

--Figure 4: I would have liked to see a little more data about the "numerous new potential sRNAs".

--Figures a, b and c: I am not clear what these panels, which take up quite a bit of space, are showing (they also are redundant with Supplementary Figure 5).

--Browser images: Some indication of scale/signal would be useful. For example, in Figure 4c, if the lack signal for RsmY in PA14 is because the RNA is "slightly less abundant in PA14", what are the levels of RsmY?

--I appreciate that the authors used different colors for PAO1, PA14 and IHMA87 throughout, but the yellow color used for IHMA87 is not optimal. For example, the peaks are hard to see in Figure 6a.

3. More minor editorial suggestions:

--The following statements did not make sense to me.

Lines 117-118: "there was a 3- to 10-fold stronger overlap in transcripts between transition and stationary phases"

Lines 161-163: "Among the strain-specific targets, there were few transcripts encoding..."

Lines 188-190: "Interestingly, most of the differences between strains stemmed from the conservation of sRNAs themselves..."

Line 422: "reflects phenotypic readouts"

--Suggestions:

Line 374: Replace "verify" with "test."

--Typographical errors:

Line 427: "investigated in more details"

In general, the text should be re-read to eliminate non-standard or ambiguous phrasing.

Gisela Storz

Reviewer #2:

Remarks to the Author:

The manuscript « The core and accessory Hfq interactomes across *Pseudomonas aeruginosa* lineages" by Trouillon et al. deals with the very interesting and important question of how the Hfq-dependent regulon changes within the limits of one fast-evolving species represented by thousands of ecologically dissimilar isolates with quite different genomes. The importance of Hfq is well established in a number of model species, and it is usually tacitly accepted that the regulatory phenomena observed in one studied strain hold for the entire species, if not the genus or even the

family, provided all involved players are conserved. However, this assumption has been seldom tested experimentally, and even less so in a genome-wide comparative study like the one presented here.

P. aeruginosa seems to be an excellent model to tackle this question. The authors explored the Hfq interactomes of three strains representative of distinct clades and found that they differ in part due to variable accessory genomes but also at the level of the core genome, some conserved RNAs demonstrating strain-specific Hfq-binding patterns. With a number of examples, the authors report on new sRNAs and other Hfq ligands, thereby complementing earlier transcriptomic and interactomic studies (e.g. Wurtzel et al., *PLoS Pathog* 2012; Zhang et al., *Mol Microbiol* 2017; Kambara et al., *Cell Rep* 2018; Chihara et al. *mSystems* 2019). The presented data brought the authors to two interesting, however contentious (see below), conclusions: (i) the core Hfq interactome varies between strains and is independent of the strain-specific RNA abundances (i.e. there must be differential Hfq affinity towards the same transcripts in different strains); (ii) the three strains have different, context-dependent Hfq-mediated regulons. Finally, using rGRIL-seq, the authors convincingly identify the previously reported RhIS as an Hfq-dependent base-pairing sRNA positively regulating the expression of *vfr*, thus revealing a new link between quorum sensing and virulence.

This study is well-written and contains all information required for replication. It is definitely very important for the field of bacterial RNA biology and will be of interest for broader readership interested in the evolution and plasticity of posttranscriptional regulatory networks. However, a few essential points will have to be addressed prior to publication.

Major points:

1. The authors on many occasions interpret the association of specific RNAs with Hfq as evidence for “a new conserved role for Hfq”, “growth-dependent major roles of Hfq... that affect globally the core and accessory metabolisms as well as bacterial communication systems”, “a novel Hfq-dependent regulatory mechanism”, or “the central role of Hfq” in the corresponding pathways. However, binding does not equal regulation: Hfq can interact with transcripts without noticeably affecting their stability or translation. Making the point that the three strains have substantially different Hfq-dependent regulons requires not only interactomic data but also the analysis of gene expression changes between WT and the corresponding *hfq* deletion strains. This would permit to evaluate the genuine scope of Hfq-mediated regulation in each strain and connect the rather descriptive interactome analysis to biologically relevant functional outputs.
2. The authors consider that the differences between the core Hfq interactomes of the three strains cannot be explained by the differential abundance of RNA ligands (i.e. gene expression). This conclusion seems to be at odds with earlier observations in *P. aeruginosa* (Chihara et al. *mSystems* 2019) and other species, where Hfq binding correlated well with the expression levels of ligands and the Hfq interactome was shown to undergo facile condition-specific reorganisation via RNA cycling, largely reflecting the transcriptional state of the cell. I consider that this authors’ conclusion is not supported by the data presented in Supplementary Fig. 6. Pearson correlation coefficients of 0.7-0.8 mean quite substantial differences between transcript levels (as indeed seen on the panel b); and when one looks at the panel c, it is clear that differentially enriched genes are much less correlated between strains (i.e. they are differentially abundant). To robustly evaluate whether enrichments are abundance-driven, I propose a simple statistical procedure. The authors can easily test the hypothesis that the genes specifically bound by Hfq in the strain A but not in the strain B tend to be more highly expressed in A than in B, as compared to all shared genes. This can be done by plotting the cumulative distributions of the corresponding fold-changes between WT samples (as proxies for gene expression) and running a Kolmogorov-Smirnov test (expression levels of differentially enriched genes vs all genes). This comparison should be done for all strain combinations in each growth phase. It may also be informative to compare different growth phases for the same strain to see whether the Hfq interactome follows the transcriptomic state over growth.
3. Quite strong conclusions are drawn from only two replicates per condition. This limits the statistical power of the study and exposes the datasets to potential confounding issues (e.g. eventual batch effects due to different sequencing platforms). Can the authors show how similar the replicates actually are? Even though DESeq2 can handle two replicates, this is a somewhat artificial workaround. Can the authors further corroborate their findings by including northern blots or RT-qPCR measurements for selected transcripts (e.g. those of particular interest, shown on Fig. 3g, 4c, and 6)?

Minor points:

1. RhIS binds the *vfr* 5'-UTR far upstream of the translation initiation site. Can the authors propose a potential mechanism of translation activation in this case (structural rearrangement? RsmA sequestration?..)
2. Can Hfq bind to the *vfr* mRNA in the absence of RhIS?
3. L. 42: the references 9 and 10 seem to be irrelevant in this context.
4. Ll. 147-149: the sentence "A number of targets..." is a bit confusing at the first read ("...shared between each pair and Hfq..."). Consider using extra punctuation to make it plain.
5. L. 154: "confirming" is hardly suitable in this context since the hfq autoregulation was initially described in a different species. "Similar to" or "paralleling" are more appropriate.
6. L. 297: the reference to Supplementary Fig. 6a is not appropriate.
7. L. 435: "was used for".
8. L. 502: specify the "4X loading buffer".
9. L. 523: "supernatants".
10. L. 583: "chimeric RNAs were precipitated".
11. Ll. 594 and 596: "coordination numbers" is a term of coordination chemistry. Better use "genomic coordinates".
12. There is no reference to Fig. 2f in the text. Regarding this figure, the phases in which the GO term enrichments are observed are quite different between strains. What may be the reason for this? Can the authors discuss this point in further detail?
13. Fig. 6: From the sequencing data it must be possible to establish whether Hfq interacts with mature crRNAs or an unprocessed precursor. Can the authors check this? Is the bound sequence conserved in PA14 or is it IHMA87-specific? Can the authors hypothesise why exactly these spacers are bound by Hfq (any sequence determinants?)
14. Figs 7a-c and S5 seem to be redundant.
15. Supplementary Fig. 1: SD should not be calculated for $n = 2$.
16. Supplementary Fig. 2: insert the reference in the caption.
17. Certain references require editing (e.g. #18))

Reviewer #3:

Remarks to the Author:

Summary:

Genome variability of *Pseudomonas aeruginosa* is a well-documented fact since many genome sequences have been completed and are freely available for further analyses. Phenotypic variability between strains is also a well-known phenomenon and the mechanisms behind this is still not very well-worked out since genome sequences are insufficient to determine transcriptional and post-transcriptional regulations, which *P. aeruginosa* employs intricate and complex pathways with multiple global regulators. The authors of this manuscript focus on one such regulator: Hfq. Using various next generation sequence tool platforms, the manuscript concludes that Hfq-global RNA interactomes show more variability between strains that are evolutionarily farther apart, with notable differences in virulence gene regulations, while revealing some novel elements of the Hfq regulons such as CRISPR crRNAs and other regulatory ncRNAs.

Major points:

1. The authors do comprehensive analyses that depend heavily on bioinformatics-assisted next generation sequencing technologies such as RIP-seq and rGRIL-seq, but with the exception of the experiments outlined in Fig. 8, I do not see any evidence of follow-up experiments on biochemical/molecular biology levels, nor ultimately on microbiological levels that relate to the big pictures that involve phenotypic outputs. Where are gel shift experiments to corroborate their high throughput results? I see qRT-PCR is featured in the Methods (lines 510-515), but it is not a featured experiment in the figures. Western blots against T3SS targets? Virulence/infection experiments to phenotypically support the regulatory model?
2. There is a fundamental issue with the authors making broad speculations that amounts to Hfq-

related mechanisms pretty much must involve sRNAs. Despite acknowledging that there is precedence in sRNA-independent regulation (e.g. line 38), the authors do not mention that there have been many examples where Hfq binds independently to non-sRNAs (such as mRNAs) and Hfq-binding consensus are also well-known. Particularly given that *P. aeruginosa* has extremely intricate regulatory pathways with multiple branching and convergent points, with our knowledge extending to only the tip of the iceberg, I do not see how the authors can make such blanket assumption that automatically leads their logic towards the existence of sRNA when Hfq is involved. I raise the following points to highlight this, and these are not the exclusive list:

- line 13 (mediating sRNAs and target mRNA is not the only function of Hfq)
- line 40 (how does one show that Hfq is a "primarily" a sRNA-mRNA matchmaker when there is still so much to uncover?)
- line 245 (the usage of "potentially" helps, but the authors immediately run to the logic of sRNA)
- line 286 (though it worked out that way with the description of how RhIS affects *vfr* translation, here again, the logic goes straight to the presence of sRNA which really needs to be modified with how it is written so that other options are not immediately ruled out for no reason)

The authors make no mention of the possibilities that Hfq directly bind to the leader sequence adjacent to the ORF. I believe the authors will need to run a genome-wide promoter (and terminator) finder analyses in conjunction to properly conclude whether intergenic regions are truly candidates of sRNAs or not. As the authors are very aware with their very example *vfr*, many genes have extended regions of mRNA that are between the transcriptional and translational starts. It is wrong to simply use the characterisation of "intergenic region" (i.e. in between annotated ORFs) to even suggest that they are sRNAs (some might be yes, but many might not be and one does not know because the authors never ran these analyses) to rule one or the other out.

3. Hfq has several known affinity sequences that they bind to: ARN repeats (distal), U-rich (proximal), and UA-rich (rim) sequences. These critical features are not even mentioned in the manuscript. This is important because with the vast amount of data that were collected by the authors, it would not be difficult to make some kind of relationships of their finding with any of the three known affinities. These data will provide additional strength to the Hfq-binding sites identified by RIP-Seq. The only time we can closely see Hfq binding sites clearly is in Supplementary Fig. 4 for Hfq-binding on *vfr* mRNA. Despite showing the potential Hfq-binding sites as proposed by Irie *et al*. in green boxes, there are no discussions regarding this in comparison to the new data presented anywhere in this manuscript. Again, without substantial biochemical and/or molecular biology experiments, the authors are too limited.

4. As PAO1 and PA14 are frequently used laboratory strains, and represent separate evolutionary clusters, the selection of these two are appropriate. The question is the specific selection of the relatively obscure IHMA87 strain (this genome sequence is not even archived on pseudomonas.com the last I checked). If this strain belonged to the same cluster as PA7, and PA7 is more commonly studied, why did the authors not select PA7? Furthermore, there are other studies that indicate that PAK is an outlier isolate of *P. aeruginosa* (Wiehlmann L *et al*., PNAS 2007), which for years, the corresponding author Professor Burkhard Tümmler who the Lory Lab has collaborated in the past, have frequently presented (e.g. featured in *Pseudomonas* 2007 meeting review written by Goldberg JB *et al*., Journal of Bacteriology 2008). Trouillon *et al*. does not reconcile with these, and other previous phylogenetic analyses that are published. It is strange that PAK, a very frequent strain studied in the Lory Lab, is also not included in the in-depth analyses in this manuscript.

Medium level points:

1. Fig. 8d: Why is the value of mutant RhIS+compensatory *vfr* mutant without arabinose lower than the others? Would you not expect the value to be identical to the other white bars? Please explain. This is particularly important considering the + arabinose counterpart is indeed higher than no arabinose, but the number is approximately equal to the other white bars. We therefore cannot make conclusive judgements of whether the no arabinose control is artefactual and + arabinose is not actually working. Had this grey bar value matched approximately 400 Miller Units, this would have been better, but this is not case.

2. Line 425: "strain-specific regulatory role of the Hfq target, as illustrated for Vfr." As far as I can see from Fig. 3f, it appears Hfq binds to *vfr* **regardless** of strains. It's possible that I am getting confused with the wording. Please either clarify or re-write.

Minor points:

1. Line 40: "major human pathogen" is rather hyperbolic and over-blowing/self-promoting. Perhaps a simple "opportunistic" will be more descriptive and appropriate?

2. Line 102: Did the authors mean to say "PA7-like" instead of "IHMA87-like"?

3. Figures out of sequence: Fig. 3a appears on line 149, followed by 3d in line 150, then 3e, 3f, 3b, 3c, and 3g. Please re-arrange in order.

4. Line 193: Prf2 → PrfF2

5. Line 198: sRNAs → sRNA

6. Line 210: *prf1-2* → *prfF1-2*

7. Fig. 4d: Please include RsmY in the y-axis label since this graph is different from all the others that have gene names indicated in the graphs. This one does not and "RPM" is insufficient.

8. Throughout the manuscript: "5' UTR" is an obsolete term in bacterial genetics since many transcribed portion of mRNAs 5' to ORF are translated (as part of how they are regulated). *trp* operon is one of the most famous examples of this. Therefore, post-transcriptional regulation experts have switched to "5' leader sequence" and progressing slowly within the field. This manuscript deals heavily with post-transcription field, coming from a high-impact laboratory from high-impact university, so I implore that you serve as an example for those who are still not aware of this more accurate description of this region of the mRNA.

9. Line 359: "directly" - as stated earlier, the authors did not perform biochemical assays to prove that RhIS is indeed **directly** regulating *vfr*. Without any mechanistic insights, this is a stretch at best.

10. Everywhere in the manuscript: the "L" of L-arabinose should be smaller font size.

11. Everywhere in the manuscript: Did the authors use OD (optical density) or A (absorbance)? Please read your equipment manuals to clarify whether it is OD or A. In my experience, most modern equipment are reading absorbance but the authors will need to check for their own equipment.

12. Line 624: "Julian Trouillon" → JT (like all other authors)

13. Line 677: The title of the article is repeated.

14. Supplemental Fig. 1a: what is the *n* of this growth curve experiment?

15. Supplemental Fig. 1c: PrfA in y-axis → PrfF1?

16. Supplemental Fig. 2 legend (line 13): Did the authors mean to insert something in place of "(ref)"?

17. Supplemental Fig. 4 legend (line 33): Multialin → Multalin

18. While the manuscript is overall written to be very clear with excellent English, the Methods section is ever so slightly lower in quality. Please revise. I list below some examples of what I detected:

a. Line 496: HFQ → Hfq

- b. Line 499: Arabinose → arabinose
- c. Line 500: is it 0.2% w/v?
- d. Line 505: Are anti-Vfr and anti-FliC rabbit antibodies?
- e. Line 525: What is the exact ratio between phenol/chloroform/isoamyl alcohol (and the P, C, and I do not have to be capitalised)
- f. Line 531: DNase → DNase
- g. Line 532: Thermofisher → Thermo Fisher
- h. Line 533: What is the exact ratio between phenol/chloroform (and the P and C do not have to be capitalised)
- i. Line 568: Luria-Bertani is incorrect. The authors have already correctly used "Lysogeny broth" as LB in line 438.
- j. Lines 574, 577: space between number and g
- k. Line 576: utsing → using
- l. Lines 500, 578, 584: remove space between number and %
- m. Lines 583, 584: remove space between number and ^oC

REVIEWER COMMENTS

Reviewer #1 (Remarks to the Author):

Trouillon et al. examined the conservation of small, regulatory RNAs (sRNAs) and RNA binding proteins (RBPs) across *Pseudomonas aeruginosa* lineages and assayed the interactomes of the RNA binding protein Hfq in laboratory strains PAO1, PA14 and IHMA87. The authors then examined the interaction of the mRNA encoding the global regulator Vfr with the RhIS sRNA in more detail. This comprehensive study represents a significant amount of work and is an important contribution to the field, providing useful datasets for investigators interested in sRNA and RBP evolution. Most of my comments are editorial in nature.

1. Given the amount of data presented in this study, the text of the first results sections should be tightened to avoid having too many lists of numbers and to better highlight the main conclusions. I also would have liked more analysis of the factors that contribute to the strain differences in the Hfq interactomes since the differences “cannot be explained by different RNA abundances”. All or parts of Supplementary Figure 6 should be included in the main text.

As detailed in our answer to Reviewer 2, who suggested complementary analyses to Figure S6, we provide new statistical evidence to support the claim that the differences in RNA abundance are not sufficient to explain all differences observed in RIP-seq enrichment. As we agree with Reviewers 1 and 2 that this claim is an important result of this article, we moved (and modified) the corresponding text from the discussion to the results section where we also comment on the corresponding new figures (see response to reviewer 2 for more details).

We notably added clear explanations of our views on that point, supported by these new comprehensive analyses in support of our claims. Briefly, we believe that two main factors can lead to differences in binding to Hfq for conserved transcripts; i) difference in RNA abundance, ii) difference in binding affinity. While we could assess the importance of RNA abundance, we cannot really investigate differences in binding affinities with our approach. Binding affinity could be changed by differences at the RNA binding site or the presence/absence of other co-interactants (such as sRNAs or other proteins). Both mechanisms would require different approaches than RIP-seq to study, since it cannot identify precise binding sites (as discussed below) or the role of other RNAs/proteins.

2. The authors should similarly be judicious about what they include in their figures.

--Figure 2: I suggest moving the heatmaps (for which the shades are difficult to distinguish) to supplementary information and substituting browser images documenting examples of changes across growth

Done, see Figure 2g, 2h, 2i.

--Figure 3: Panels d and e are not very informative

These panels were moved in the figure to make more sense in the order the data is presented.

--Figure 4: I would have liked to see a little more data about the “numerous new potential sRNAs”.

We preferred not to extensively comment on most new sRNAs since they are potential and not yet confirmed. However, we are commenting more on the 7 high-confidence sRNAs, and added a comment on a detected TSS for one of them.

--Figures a, b and c: I am not clear what these panels, which take up quite a bit of space, are showing (they also are redundant with Supplementary Figure 5).

Supplementary Figure 5 was removed as it was indeed redundant. We believe showing the entire coverage in the main figure is relevant as it shows that there is only one strong hit across the chromosomes.

--Browser images: Some indication of scale/signal would be useful. For example, in Figure 4c, if the lack signal for RsmY in PA14 is because the RNA is “slightly less abundant in PA14”, what are the levels of RsmY?

RsmY levels are shown in Figure 4d.

--I appreciate that the authors used different colors for PAO1, PA14 and IHMA87 throughout, but the yellow color used for IHMA87 is not optimal. For example, the peaks are hard to see in Figure 6a.

Figure 6a has a zoomed section on the peaks to see them better. The yellow color doesn't seem to be an issue in any other figures.

3. More minor editorial suggestions:

--The following statements did not make sense to me.

Lines 117-118: “there was a 3- to 10-fold stronger overlap in transcripts between transition and stationary phases” (line 117-118)

Lines 161-163: “Among the strain-specific targets, there were few transcripts encoding...” (line 171-173)

Lines 188-190: “Interestingly, most of the differences between strains stemmed from the conservation of sRNAs themselves...” (line 200-202)

Line 422: “reflects phenotypic readouts” (line 463)

--Suggestions:

Line 374: Replace “verify” with “test (line 302).

--Typographical errors:

Line 427: “investigated in more details” (line 473)

In general, the text should be re-read to eliminate non-standard or ambiguous phrasing.

We addressed all above editorial suggestions and proof-read and corrected the text.

Gisela Storz

Reviewer #2 (Remarks to the Author):

The manuscript « The core and accessory Hfq interactomes across *Pseudomonas aeruginosa* lineages” by Trouillon et al. deals with the very interesting and important question of how the Hfq-dependent regulon changes within the limits of one fast-evolving species represented by thousands of ecologically dissimilar isolates with quite different genomes. The importance of Hfq is well established in a number of model species, and it is usually tacitly accepted that the regulatory phenomena observed in one studied strain hold for the entire species, if not the genus or even the family, provided all involved players are conserved. However, this assumption has been seldom tested experimentally, and even less so in a genome-wide comparative study like the one presented here.

P. aeruginosa seems to be an excellent model to tackle this question. The authors explored the Hfq interactomes of three strains representative of distinct clades and found that they differ in part due to variable accessory genomes but also at the level of the core genome, some conserved RNAs demonstrating strain-specific Hfq-binding patterns. With a number of examples, the authors report on new sRNAs and other Hfq ligands, thereby complementing earlier transcriptomic and interactomic studies (e.g. Wurtzel et al., PLoS Pathog 2012; Zhang et al., Mol Microbiol 2017; Kambara et al., Cell Rep 2018; Chihara et al. mSystems 2019). The presented data brought the authors to two interesting, however contentious (see below), conclusions: (i) the core Hfq interactome varies between strains and is independent of the strain-specific RNA abundances (i.e. there must be differential Hfq affinity towards the same transcripts in different strains); (ii) the three strains have different, context-dependent Hfq-mediated regulons.

Finally, using rGRIL-seq, the authors convincingly identify the previously reported RhIS as an Hfq-dependent base-pairing sRNA positively regulating the expression of *vfr*, thus revealing a new link between quorum sensing and virulence.

This study is well-written and contains all information required for replication. It is definitely very important for the field of bacterial RNA biology and will be of interest for broader readership interested in the evolution and plasticity of posttranscriptional regulatory networks. However, a few essential points will have to be addressed prior to publication.

Major points:

1. The authors on many occasions interpret the association of specific RNAs with Hfq as evidence for “a new conserved role for Hfq”, “growth-dependent major roles of Hfq... that affect globally the core and accessory metabolisms as well as bacterial communication systems”, “a novel Hfq-dependent regulatory mechanism”, or “the central role of Hfq” in the corresponding pathways. However, binding does not equal regulation: Hfq can interact with transcripts without noticeably affecting their stability or translation. Making the point that the three strains have substantially different Hfq-dependent regulons requires not only interactomic data but also the analysis of gene expression changes between WT and the corresponding *hfq* deletion strains. This would permit to evaluate the genuine scope of Hfq-mediated regulation in each strain and connect the rather descriptive interactome analysis to biologically relevant functional outputs.

We agree with reviewer 2 that some sentences in the text might wrongly suggest that binding necessarily implies regulation, although this is probably true in many cases. However, it indeed cannot be directly implied without actually assessing the effect on RNA stability or translation. To address this point, that we believe to be more editorial in nature, we adjusted/rephrased the corresponding claims all along the text.

While it would indeed be interesting to assess changes in gene expression (using RNA-seq for instance), we believe that our claim and the main message of this article (existence of important intra-species differences in the interactomes of RNA-binding proteins) are fully supported by our analysis.

2. The authors consider that the differences between the core Hfq interactomes of the three strains cannot be explained by the differential abundance of RNA ligands (i.e. gene expression). This conclusion seems to be at odds with earlier observations in *P. aeruginosa* (Chihara et al. mSystems 2019) and other species, where Hfq binding correlated well with the expression levels of ligands and the Hfq interactome was shown to undergo facile condition-specific reorganisation via RNA cycling, largely reflecting the transcriptional state of the cell. I consider that this authors' conclusion is not supported by the data presented in Supplementary Fig. 6. Pearson correlation coefficients of 0.7-0.8 mean quite substantial differences between transcript levels (as indeed seen on the panel b); and when one looks at the panel c, it is clear that

differentially enriched genes are much less correlated between strains (i.e. they are differentially abundant).

It is clear from the comments of the reviewers that we pushed the claim that “difference in interactions cannot be explained by differential RNA abundances” too strongly in the text. This was not our intention (as shown by the few examples where we actually claim that some of the observed differences are in fact due to different RNA abundances - Fig. 4d & Fig. S7). Our intention was more to highlight the fact that - as opposed to what could be expected - many (but not all) of the differences are not explained by different RNA abundances. We agree with the reviewers that our phrasing is sometimes wrong and/or too strong; we consequently made several editorial changes to the text to soften that claim.

To robustly evaluate whether enrichments are abundance-driven, I propose a simple statistical procedure. The authors can easily test the hypothesis that the genes specifically bound by Hfq in the strain A but not in the strain B tend to be more highly expressed in A than in B, as compared to all shared genes. This can be done by plotting the cumulative distributions of the corresponding fold-changes between WT samples (as proxies for gene expression) and running a Kolmogorov-Smirnov test (expression levels of differentially enriched genes vs all genes). This comparison should be done for all strain combinations in each growth phase. It may also be informative to compare different growth phases for the same strain to see whether the Hfq interactome follows the transcriptomic state over growth.

We thank the reviewer 2 for this very important suggestion. As suggested, we extracted the lists of differentially enriched transcripts for all 18 strain combinations in each growth phase and performed Kolmogorov-Smirnov tests to assess whether the distributions of abundance fold changes (from WT samples) were higher for differentially (positively) enriched genes than for all genes. Out of the 18 tested combinations, the abundance fold-change distributions of differentially enriched transcripts were found significantly higher (with $p < 0.01$) in only three cases. We report this analysis in a new figure (Figure 7) along with the previous figure S6, as suggested by reviewers 1 & 2, where we show the test results along with density plots of all analyzed distributions to allow for visual observation of this claim. We believe that the results of this analysis corroborate and strengthen our claim that most of the observed differences are not due to abundance differences.

Fig. 7. The effect of RNA abundance on inter-strain enrichment differences. (a) Heatmap of Pearson correlation coefficient from RNA abundance rank-matching analysis between all strains and growth phases. (b) Comparison of RNA abundance ranks for all shared genes between IHMA87 and PAO1 in the stationary phase. (c) Comparison of RNA abundance ranks for only genes differentially enriched between IHMA87 and PAO1 in the stationary phase. (d) Density plots of abundance fold change distributions for all 18 strain combinations in each growth phase. Kolmogorov-Smirnov tests were used to assess if abundance fold changes were significantly higher for differentially enriched genes (red) against all conserved genes (blue). p -values are reported on each corresponding plot (green: <0.01 , orange: <0.05 , red: >0.05) along with the number of differentially enriched genes (n).

3. Quite strong conclusions are drawn from only two replicates per condition. This limits the statistical power of the study and exposes the datasets to potential confounding issues (e.g. eventual batch effects due to different sequencing platforms). Can the authors show how similar the replicates actually are? Even though DESeq2 can handle two replicates, this is a somewhat artificial workaround. Can the authors further corroborate their findings by including northern blots or RT-qPCR measurements for selected transcripts (e.g. those of particular interest, shown on Fig. 3g, 4c, and 6)?

To assess potential differences between our biological replicates, we computed the Pearson correlation of feature counts between all sample pairs in each growth condition. Notably, this analysis showed that all replicate pairs are highly similar (Pearson correlation >0.9). This also showed that Hfq-3xFLAG samples were also, as expected, much closer to each other than to control WT samples. We report this analysis in a new supplementary figure (Supplementary Fig. 2).

Reviewer 2 also suggests that we support some of the highlighted results with confirmation experiments. This is also something that reviewer 3 suggested, to avoid repetition, we answer to that suggestion in our answer to reviewer 3 (major point 1).

Supplemental Fig. 2. Replicates correlation analysis. (a-c) Hierarchical clustering and heatmap of Pearson correlation coefficient between all biological replicates based on features counts from genes conserved in all three strains for exponential (a), transition (b) and stationary (c) phases. (d) Pearson correlation coefficient between paired biological replicates.

Minor points:

1. RhIS binds the *vfr* 5'-UTR far upstream of the translation initiation site. Can the authors propose a potential mechanism of translation activation in this case (structural rearrangement? RsmA sequestration?..)

Based on our predicted base-pairing model and complementation analysis, we infer that RhIS interacts with the complementary sequence of *vfr* 5' UTR and this interaction seems to be required for activation of the *vfr* translation. As shown in Fig. 9d, the mutation of RhIS abolishes the

activation while the compensator mutation in the interaction site of *vfr* mRNA restores the activation. As the reviewer mentioned, it is possible that RhIS could induce a rearrangement of the structure of *vfr* mRNA leading the Hfq binding and opening the Shine-Dalgarno sequence as suggested in the previous study (Irie *et al.*, Front Microbiol., 2020). As shown in the Irie's study, the 5' UTR of *vfr* mRNA was proposed to form a stem-loop structure and it encompasses the Hfq-binding sites, RsmA-binding sites and the Shine-Dalgarno sequence. RhIS could facilitate Hfq-binding on the stem-loop structure by rearranging the structure as shown in several positive sRNA regulators (DsrA for *rpoS* regulation; Branislav *et al.* Nucleic Acid Res., 2009 and RyhB for *shiA* regulation; Karine *et al.*, Mol. Micro., 2007). It is also possible that RhIS protects the *vfr* mRNA against RNases (exo- or endo-nuclease), as shown for RydC which protects the *cfa1* mRNA against RNase E (Frohlich *et al.*, EMBO, 2013). We added these comments regarding the possible activation mechanisms in the Discussion section (line 442-448).

2. Can Hfq bind to the *vfr* mRNA in the absence of RhIS?

This point has been addressed previously (Irie *et al.*, Front Microbiol., 2020). In that study, the authors used an *in vitro* gel shift assay and Hfq was shown to bind to *vfr* mRNA without RhIS. However, the *vfr* mRNA used for this assay doesn't encompass the predicted base-pairing site of RhIS that we suggested (They used 223-nt *vfr* RNA encompassing coordinated -106 to +116 relative to the initiation codon of *vfr*, while the predicted base-pairing site of RhIS that we suggested is coordinated -139 to -129). Therefore, it remains to be determined whether or not RhIS would influence on the Hfq binding of 5' UTR of *vfr* mRNA when the assay is carried out with full length of 5' UTR of *vfr* mRNA. However, we intend to study the detailed mechanism of RhIS action on Vfr in the near future. Here, we used it as an example to highlight the rGRIL-Seq tool to differentiate Hfq binding during stabilization of sRNA-mRNA interaction from other Hfq-RNA interactions.

3. L. 42: the references 9 and 10 seem to be irrelevant in this context (line 42).

4. LI. 147-149: the sentence "A number of targets..." is a bit confusing at the first read ("...shared between each pair and Hfq..."). Consider using extra punctuation to make it plain (line 162-164).

5. L. 154: "confirming" is hardly suitable in this context since the hfq autoregulation was initially described in a different species. "Similar to" or "paralleling" are more appropriate (line 173).

6. L. 297: the reference to Supplementary Fig. 6a is not appropriate (line 351).

7. L. 435: "was used for" (line 479).

8. L. 502: specify the "4X loading buffer" (line 551).

9. L. 523: "supernatants" (line 572).

10. L. 583: "chimeric RNAs were precipitated" (line 630).

11. LI. 594 and 596: "coordination numbers" is a term of coordination chemistry. Better use "genomic coordinates" (line 637-641).

We addressed the above editorial suggestions (points 3 to 11) and corrected the text.

12. There is no reference to Fig. 2f in the text. Regarding this figure, the phases in which the GO term enrichments are observed are quite different between strains. What may be the reason for this? Can the authors discuss this point in further detail?

A reference to figure 2f was added with the corresponding text (line 147-148). Previous studies on the topic would suggest that these genes should be enriched towards transition and stationary phases. Why are they also enriched in exponential phase in the IHMA87 strain is unclear.

13. Fig. 6: From the sequencing data it must be possible to establish whether Hfq interacts with mature crRNAs or an unprocessed precursor. Can the authors check this? Is the bound sequence conserved in PA14 or is it IHMA87-specific? Can the authors hypothesise why exactly these spacers are bound by Hfq (any sequence determinants?)

The fact that the enrichment signal spans several crRNAs in the cluster suggests a binding of Hfq to the unprocessed guide RNA. We added a comment on that in the corresponding section (line 282-287). Additionally, we ran a motif search for the Hfq motif (from Chihara *et al.* 2019 - *mSystems*) on the crRNA spacer sequences from both IHMA87 and PA14 and found that one spacer in the IHMA87 cluster (and none in PA14) matched the 5 AAN repeats Hfq motif, which could explain the observed strain-specific binding of Hfq.

14. Figs 7a-c and S5 seem to be redundant.

Figure S5 has been removed.

15. Supplementary Fig. 1: SD should not be calculated for $n = 2$.

SD has been changed for s.e.m.

16. Supplementary Fig. 2: insert the reference in the caption.

This change has been made.

17. Certain references require editing (e.g. #18))

This change has been made.

Reviewer #3 (Remarks to the Author):

Summary:

Genome variability of *Pseudomonas aeruginosa* is a well-documented fact since many genome sequences have been completed and are freely available for further analyses. Phenotypic variability between strains is also a well-known phenomenon and the mechanisms behind this is still not very well-worked out since genome sequences are insufficient to determine transcriptional and post-transcriptional regulations, which *P. aeruginosa* employs intricate and complex pathways with multiple global regulators. The authors of this manuscript focus on one such regulator: Hfq. Using various next generation sequence tool platforms, the manuscript concludes that Hfq-global RNA interactomes show more variability between strains that are evolutionarily farther apart, with notable differences in virulence gene regulations, while revealing some novel elements of the Hfq regulons such as CRISPR crRNAs and other regulatory ncRNAs.

Major points:

1. The authors do comprehensive analyses that depend heavily on bioinformatics-assisted next generation sequencing technologies such as RIP-seq and rGRIL-seq, but with the exception of the experiments outlined in Fig. 8, I do not see any evidence of follow-up experiments on biochemical/molecular biology levels, nor ultimately on microbiological levels that relate to the big pictures that involve phenotypic outputs. Where are gel shift experiments to corroborate their high throughput results? I see qRT-PCR is featured in the Methods (lines 510-515), but it is not a featured experiment in the figures. Western blots against T3SS targets? Virulence/infection experiments to phenotypically support the regulatory model?

qRT-PCR is indeed featured in the methods, and corresponds to the qRT-PCR experiments shown in Figure S1.

We believe that confirmation experiments done simply to confirm the results with another method (such as doing RT-qPCR to confirm results of a transcriptome analysis) often do not bring any additional biological insights, especially now that most major NGS-based approaches have proved their value and are widely accepted as such. Also, Hfq has been widely studied and our data confirm numerous previously known results, further validating our approach. Consequently, we initially decided instead to focus on one interesting new result and to study it in depth.

For that reason, we focused on the Vfr case and performed confirmation and follow-up experiments including rGRIL-seq in 3 strains and 2 conditions to discover a new involved sRNA and assays including translational fusions and western blots with several functional mutations done at the RNA-RNA interaction sites to confirm and actually bring new information on this new interaction. We believe this represents more work than most standard NGS confirmation experiments and most importantly brings much more new biological insights.

Since confirmation experiments were asked by both reviewers 2 and 3, including demonstration of interesting phenotypes, we performed the following follow-up experiments:

To assess the phenotypic consequences of the observed interaction between RhIS and vfr mRNA, we created a translational fusion for one major Vfr target (*lasR*, Albus *et al.* J Bacteriol. 1997) and monitored the effect of RhIS on its expression. We observed an increase in the expression of the *lasR-lacZ* reporter upon expression of RhIS, as expected from the observed effect of RhIS on Vfr.

This new experiment was added as an additional panel to Figure 9, commented in the main text and described in the Methods section.

Figure 9f: (f) β -galactosidase activity from the P_{lasR} -*lacZ* transcriptional fusion in rhIS mutant strain containing empty vector (pKH6) or RhIS expression (pKH6-RhIS). Data are shown as mean \pm SD for three biological replicates. Statistical comparisons were performed using Student's t-test. *** $P \leq 0.001$.

Additionally, as requested, we also performed an infection experiment, to confirm our regulatory model. We saw a small decrease of cytotoxicity towards epithelial cells of the RhIS mutants, as would be expected from a decreased Vfr activity/amount. However, due to our current lab situations (explained below), we could not perform enough replicates for this experiment and thus do not feel comfortable adding it to the manuscript. The phenotype seems mild and we expect that a lot of factors influence this result in these *ex vivo* conditions, which would require a more in depth analysis of the pathogenicity phenotype of this mutant, for which we don't have enough time and resources.

Kinetics of bacterial cytotoxicity on epithelial cells. A549 epithelial cells were infected with indicated PA14 strains at a multiplicity of infection (MOI) of 10 in the presence of propidium iodide (PI). PI incorporation, which reflects membrane permeabilization, was monitored every 10 min. Data are represented as mean \pm s.e.m. from three independent experiments.

While we tried our best to address the reviewer's comments, our current lab situations made experimental work very hard to perform. Indeed, both authors that have performed the experiments of this study (JT and KH) have now left the Lory and Attrée labs and are not able to work with level 2 pathogens (such as *P. aeruginosa*) in their current labs. Also, the Lory lab, where the study was performed, is closing down and no experimental members remain that could help perform the asked experiments. The Attrée lab is not fitted for this type of RNA work (which is why JT moved to the Lory lab for this project) and is additionally currently moving between institutes and locations, making any potential experiment very hard. We hope this will be taken into account when assessing our answer, as we did do our best over the last months and still managed to perform the two experiments presented above, and as we otherwise addressed all computational and editorial concerns.

2. There is a fundamental issue with the authors making broad speculations that amounts to Hfq-related mechanisms pretty much must involve sRNAs. Despite acknowledging that there is precedence in sRNA-independent regulation (e.g. line 38), the authors do not mention that there have been many examples where Hfq binds independently to non-sRNAs (such as mRNAs) and Hfq-binding consensus are also well-known. Particularly given that *P. aeruginosa* has extremely intricate regulatory pathways with multiple branching and convergent points, with our knowledge extending to only the tip of the iceberg, I do not see how the authors can make such blanket assumption that automatically leads their logic towards the existence of sRNA when Hfq is involved. I raise the following points to highlight this, and these are not the exclusive list:

a. line 13 (mediating sRNAs and target mRNA is not the only function of Hfq) (line 13)

b. line 40 (how does one show that Hfq is a "primarily" a sRNA-mRNA matchmaker when there is still so much to uncover?) (line 41-43)

c. line 245 (the usage of "potentially" helps, but the authors immediately run to the logic of sRNA) (line 257)

d. line 286 (thought it worked out that way with the description of how RhlS affects *vfr* translation, here again, the logic goes straight to the presence of sRNA which really needs to be modified with how it is written so that other options are not immediately ruled out for no reason)

We agree with reviewer 3 that we might have overstated the sRNA-mRNA matchmaking role of Hfq relative to its other known roles. Therefore, we accordingly modified numerous statements, including all above-mentioned sentences (line 339-340).

The authors make no mention of the possibilities that Hfq directly bind to the leader sequence adjacent to the ORF. I believe the authors will need to run a genome-wide promoter (and terminator) finder analyses in conjunction to properly conclude whether intergenic regions are truly candidates of sRNAs or not. As the authors are very aware with their very example *vfr*, many genes have extended regions of mRNA that are between the transcriptional and translational starts. It is wrong to simply use the characterisation of "intergenic region" (i.e. in between annotated ORFs) to even suggest that they are sRNAs (some might be yes, but many might not be and one does not know because the authors never ran these analyses) to rule one or the other out.

As mentioned below, unlike CLIP-seq, RIP-seq mostly enriches full-length transcripts, and not only the bound region. For that reason, whether Hfq binds to untranslated regions or coding regions of an RNA, the whole transcript is usually enriched. This is seen in virtually all coverage figures (and basically everywhere on the genome), with enrichment encompassing more than just the coding region. Therefore, unlike CLIP-seq where the unbound parts of the transcripts are purposely shortened and degraded, RIP-seq does not allow the prediction of the binding location on the RNA. There can sometimes be some local enrichment that could be attributed to protection of the protein-bound RNA from degradation, but these are usually minor and not predictable.

This technical aspect has the disadvantage of preventing the identification of the precise binding location on the mRNA, but on the other hand makes the attribution of enrichment to transcripts easier, which basically do not necessarily need to have identified promoters and terminators for analysis, as needed for CLIP-seq or as suggested here by Reviewer 3. For this reason, RIP-seq data is classically analyzed by looking at differential enrichment (DE) on coding regions or annotated sRNAs only (Li *et al.*, 2018; Saliba *et al.*, 2017; Boudry *et al.*, 2020; Li *et al.*, 2021 ...).

Here, in addition to the classical DE analysis, we complemented it with peak calling to try to capture additional enrichment in unannotated regions. We used a stringent selection for the identification of peaks representing potential new sRNAs by first selecting peaks that are strictly intergenic (i.e. that are entirely found in one intergenic region and not overlapping with any coding regions around, even by just 1bp). In the light of the technical aspect of RIP-seq explained above, this already means that these peaks are most probably not from mRNAs corresponding to the neighboring genes. Additionally, we further searched for known sRNAs motifs in these peaks to

identify a small group (7) of high confidence peaks representing potential sRNAs (Supplementary Fig. 3).

Although we believe this is enough to suggest that these 7 peaks potentially correspond to new sRNAs, we looked at experimentally-determined TSSs and terminator sequences (in PAO1 and PA14, no data for IHMA87) for the neighboring genes of the 3 proposed potential new sRNAs in these strains, and there was no overlap with the predicted sRNA sequences. Additionally, the potential new sRNA peak in PA14 actually fits with a previously identified TSS that was not associated with any transcript so far and predicted to correspond to an unknown intergenic sRNA (TSS found 5bp upstream of start of intergenic peak, correct orientation) (Wurtzel *et al.*, PLoS Pathog., 2012). This information has been added to the text (line 218-226).

3. Hfq has several known affinity sequences that they bind to: ARN repeats (distal), U-rich (proximal), and UA-rich (rim) sequences. These critical features are not even mentioned in the manuscript. This is important because with the vast amount of data that were collected by the authors, it would not be difficult to make some kind of relationships of their finding with any of the three known affinities. These data will provide additional strength to the Hfq-binding sites identified by RIP-Seq. The only time we can closely see Hfq binding sites clearly is in Supplementary Fig. 4 for Hfq-binding on *vfr* mRNA. Despite showing the potential Hfq-binding sites as proposed by Irie *et al.* in green boxes, there are no discussions regarding this in comparison to the new data presented anywhere in this manuscript. Again, without substantial biochemical and/or molecular biology experiments, the authors are too limited.

It would indeed be interesting to get a view of Hfq affinity sequences in our results. However, as mentioned in the text and in our answer above, one of the features of the RIP-seq method is the enrichment of full-length transcripts, unlike CLIP-seq in which RNAs are trimmed around bound proteins, therefore allowing the identification of binding motifs in relatively short sequences. Since RIP-seq doesn't give information on where an RNA binding protein binds on the transcripts and enriches mostly long sequences, such analysis is not possible. For this reason, even though we agree that it would have been nice information to have, we purposely did not comment much on that point, since it is beyond our method's limits.

4. As PAO1 and PA14 are frequently used laboratory strains, and represent separate evolutionary clusters, the selection of these two are appropriate. The question is the specific selection of the relatively obscure IHMA87 strain (this genome sequence is not even archived on pseudomonas.com the last I checked). If this strain belonged to the same cluster as PA7, and PA7 is more commonly studied, why did the authors not select PA7? Furthermore, there are other studies that indicate that PAK is an outlier isolate of *P. aeruginosa* (Wiehlmann L *et al.*, PNAS 2007), which for years, the corresponding author Professor Burkhard Tümmler who the Lory Lab has collaborated in the past, have frequently presented (e.g. featured in *Pseudomonas* 2007 meeting review written by Goldberg JB *et al.*, Journal of Bacteriology 2008). Trouillon *et al.* does not reconcile with these, and other previous phylogenetic analyses

that are published. It is strange that PAK, a very frequent strain studied in the Lory Lab, is also not included in the in-depth analyses in this manuscript.

The choice of strains for this study was based on their phylogenetic lineage (shown in Figure 1a), and as explained in the text, in order to reflect a broad view of the species' regulatory spectrum. It was especially important to have a strain from the PA7/IHMA87 lineage since it is quite distant from the rest of the species. For that reason, we did not think that the PAK strain was a good choice since it is in the PAO1-like lineage, and actually not very far from PAO1 in the phylogenetic tree.

Concerning the IHMA87 strain, its first incomplete assembly has been on Pseudomonas.com for 6 years, and its more recent complete assembly is also on the website since its publication in 2020. It is available under the ID "AZPAE15042", as noted in the articles that published its two assemblies (Kos *et al.*, *Antimicrob Agents Chemother*, 2015, Trouillon *et al.*, *Nucleic Acid Res.*, 2020), which we are still trying to have changed to the actual strain name, to avoid this kind of situation.

This strain has been thoroughly studied in >20 articles and is arising as the reference strain to study virulence through the lineage-specific ExlBA two-component secretion system. It is one of the only two strains (with PA7) with a completely assembled genome in the PA7-like lineage, which was essential for our analyses. Finally, it is sometimes chosen over the PA7 strain because, even though PA7 was the first sequenced strain in the lineage, it exhibits numerous features that are specific to that strain only in the lineage (including a strong multi-antibiotic resistance phenotype and an inactivating mutation in the global regulator Vfr). Consequently, PA7 has a very different regulatory state and also exhibits low cytotoxicity and low expression of the ExlBA secretion system, considered the major virulence factor of this lineage. These features make PA7 some kind of an outlier in its own lineage, which made us select IHMA87 as a reference strain for this lineage.

Medium level points:

1. Fig. 8d: Why is the value of mutant RhIS+compensatory *vfr* mutant without arabinose lower than the others? Would you not expect the value to be identical to the other white bars? Please explain. This is particularly important considering the + arabinose counterpart is indeed higher than no arabinose, but the number is approximately equal to the other white bars. We therefore cannot make conclusive judgements of whether the no arabinose control is artefactual and + arabinose is not actually working. Had this grey bar value matched approximately 400 Miller Units, this would have been better, but this is not case.

It is often observed that changing the 5'-UTR sequence of an mRNA influences its translation. Indeed, the substitution can also modify the RNA structure, potentially making the transcript more vulnerable/resistant to RNases or interfere with efficiency of ribosome binding, which could

explain the different basal activity. To illustrate that, we measured the activity in a strain with WT RhIS and mutated *vfr* (now added to the plot, see below; Fig. 9d), showing that the substitution induces the same basal translation reduction in this background, and no increase is seen with induction of RhIS expression. This shows that the basal level is indeed affected, probably through an unspecific mechanism due to change in 5'-UTR sequence, but is unrelated to the observed RhIS regulation, and overall further supports our conclusion.

2. Line 425: "strain-specific regulatory role of the Hfq target, as illustrated for Vfr." As far as I can see from Fig. 3f, it appears Hfq binds to *vfr* **regardless** of strains. It's possible that I am getting confused with the wording. Please either clarify or re-write.

This sentence is indeed confusing. We meant that Vfr (here the "Hfq target") has strain-specific regulatory roles (i.e. regulating T3SS in PAO1 and PA14 and ExlBA in IHMA87). We modified the sentence to make it clearer (line 471-472).

Minor points:

1. Line 40: "major human pathogen" is rather hyperbolic and over-blowing/self-promoting. Perhaps a simple "opportunistic" will be more descriptive and appropriate?

This change has been made (line 41).

2. Line 102: Did the authors mean to say "PA7-like" instead of "IHMA87-like"?

We modified this term to "PA7/IHMA87-like" all through the text (line 86, line 103), to keep the more common PA7-like form but also remind the reader that the IHMA87 strain is from that lineage.

3. Figures out of sequence: Fig. 3a appears on line 149, followed by 3d in line 150, then 3e, 3f, 3b, 3c, and 3g. Please re-arrange in order.

This change has been made.

4. Line 193: Prrf2 → PrrF2

This change has been made (line 205).

5. Line 198: sRNAs → sRNA

This change has been made (line 210).

6. Line 210: *prf1-2* → *prfF1-2*

This change has been made (line 221).

7. Fig. 4d: Please include RsmY in the y-axis label since this graph is different from all the others that have gene names indicated in the graphs. This one does not and "RPM" is insufficient.

This change has been made.

8. Throughout the manuscript: "5' UTR" is an obsolete term in bacterial genetics since many transcribed portion of mRNAs 5' to ORF are translated (as part of how they are regulated). *trp* operon is one of the most famous examples of this. Therefore, post-transcriptional regulation experts have switched to "5' leader sequence" and progressing slowly within the field. This manuscript deals heavily with post-transcription field, coming from a high-impact laboratory from high-impact university, so I implore that you serve as an example for those who are still not aware of this more accurate description of this region of the mRNA.

This change has been made.

9. Line 359: "directly" - as stated earlier, the authors did not perform biochemical assays to prove that RhIS is indeed **directly** regulating *vfr*. Without any mechanistic insights, this is a stretch at best.

We identified RhIS as the only and very strong hit in the *vfr* rGRIL-seq approach, which identifies pairs of RNAs that directly interact with each other (a requirement for the T4 RNA ligase-catalyzed reaction). We furthermore confirmed the involved regulation and identified the interaction sites on both RNAs using point mutations. We indeed did not provide *in vitro* confirmation of the interaction, however we believe that we collectively provide enough evidence to claim a direct interaction.

10. Everywhere in the manuscript

10. Everywhere in the manuscript: the "L" of L-arabinose should be smaller font size.

This change has been made.

11. Everywhere in the manuscript: Did the authors use OD (optical density) or A (absorbance)? Please read your equipment manuals to clarify whether it is OD or A. In my experience, most modern equipment are reading absorbance but the authors will need to check for their own equipment.

OD was used, as stated in the text (line 547, line 645).

12. Line 624: "Julian Trouillon" → JT (like all other authors)

This change has been made (line 667).

13. Line 677: The title of the article is repeated.

This has been modified (line 723).

14. Supplemental Fig. 1a: what is the n of this growth curve experiment?

$n=3$, this information has been added to the legend of the figure.

15. Supplemental Fig. 1c: PrrfA in y-axis → PrrF1?

This change has been made.

16. Supplemental Fig. 2 legend (line 13): Did the authors mean to insert something in place of "(ref)"?

The correct reference has been added instead (Supplementary Fig. 3).

17. Supplemental Fig. 4 legend (line 33): Multialin → Multalin

This change has been made (Supplementary Fig. 6).

18. While the manuscript is overall written to be very clear with excellent English, the Methods section is ever so slightly lower in quality. Please revise. I list below some examples of what I detected:

a. Line 496: HFQ → Hfq

This change has been made (line 545).

b. Line 499: Arabinose → arabinose

This change has been made (line 548).

c. Line 500: is it 0.2% w/v?

Yes, this information has been added (line 549).

d. Line 505: Are anti-Vfr and anti-FliC rabbit antibodies?

Yes, this information has been added (line 555).

e. Line 525: What is the exact ratio between phenol/chloroform/isoamyl alcohol (and the P, C, and I do not have to be capitalised)

The ratio was 25:24:1, the product reference has been added (line 574).

f. Line 531: DNAse → DNase

This change has been made (line 580).

g. Line 532: Thermofisher → Thermo Fisher

This change has been made (line 581).

h. Line 533: What is the exact ratio between phenol/chloroform (and the P and C do not have to be capitalised)

This change has been made (line 582).

i. Line 568: Luria-Bertani is incorrect. The authors have already correctly used "Lysogeny broth" as LB in line 438.

This change has been made (line 618).

j. Lines 574, 577: space between number and g

This change has been made (line 623, 626).

k. Line 576: utsing → using

This change has been made (line 625).

l. Lines 500, 578, 584: remove space between number and %

This change has been made (line 549, 627, 631).

m. Lines 583, 584: remove space between number and °C

This change has been made (line 630, 631).

Reviewers' Comments:

Reviewer #1:

Remarks to the Author:

The authors have made a commendable effort to address the reviewers' comments, especially given the constraints they mention. I think this is an important contribution.

Reviewer #2:

Remarks to the Author:

The authors have made a good effort to improve their original manuscript by either toning down insufficiently supported claims or performing additional experiments and analyses. This is highly appreciated, given the difficult working situation. I think this is an important and complete enough study that will promote the interest of the community in the phylogenetic and evolutionary dimensions of post-transcriptional regulation in bacteria. Below are the few remaining editorial points I spotted.

1. L. 138: "Fig. 2h", "Fig. 2i".
2. L. 143: Better would be "Our results suggest that this interaction happens...", since the authors do not show the PrrF1/PrrF2-antR interaction directly.
3. L. 283: Maybe better "... matched the known 5 AAN repeats Hfq motif..." or "...matched a known Hfq motif (5 AAN repeats)..."
4. L. 382: "Fig. 9c".
5. L. 446: "RyhB".

Alexandre Smirnov

Reviewer #3:

Remarks to the Author:

The re-submitted manuscript is a greatly improved version from the previous. There are some minor parts, mostly editorial in nature, that need to be tightened up before being accepted.

Borrowing from reviewer 2's comment 15, I do not think changing the error bar from standard deviation to standard error of the mean suddenly makes $n = 2$ statistically significant/acceptable sample size.

Response to 1 (qRT-PCR is indeed featured in the methods, and corresponds to the qRT-PCR experiments

shown in Figure S1.): This is entirely up to you and will not dictate whether the manuscript gets accepted or not, but it's worth thinking about perhaps having a separate M&M section for supplementary materials to de-clutter and abridge the already long article.

While I agree that the recent advancements of various high-throughput -omics tools are becoming more and more refined, I think it is still important for the authors to describe what their weaknesses/limitations are so that they can lead the readers into understanding why the follow up experiments were designed in certain ways and/or explain why some conclusions can't be made reliably. The authors do a good job addressing our questions as to the limitations of the tools you used (e.g. RIP-Seq) and I would like them to incorporate them more into the manuscript. I think this will only re-inforce the manuscript to an even better quality to be able to discuss the limitations to their advantages when discussing future directions.

I understand that the authors are trying to re-classify IMHA87 on pseudomonas.com, but if at the time of this manuscript currently being reviewed (and upon publication), it has not occurred yet, I believe you need to at least make a mention in your article that it is identical to AZPAE15042. There may be some people who is unaware of this connection either and it is an advantage to the readers anyway even if this switch occurs relatively soon. If not, you will be able to use this

manuscript upon publication to further convince the operators of pseudomonas.com to make the switch. All in all, I think making just one mention somewhere in the main text and/or material and methods will be a big advantage for the authors. I appreciate the information addressed to us reviewers but I'd like it reflected into the manuscript itself.

In response to the author's response (It is often observed that changing the 5'-UTR sequence of an mRNA influences its translation. Indeed, the substitution can also modify the RNA structure, potentially making the transcript more vulnerable/resistant to RNases or interfere with efficiency of ribosome binding, which could explain the different basal activity.): I absolutely agree with what the authors state here. However, it does not make sense at all, given that figure 9b shows an intact native leader sequence of vfr seemingly with only the promoter switched from native to Ptac. Therefore, according to this construct design, it should have an entirely native mRNA structure all the way up to 90 bases into the ORF, and so the sequence of mRNA should not be altered as the authors suggest here. I therefore do not think this explanation satisfies my original query.

I cannot spot that the authors have included the phenol/chloroform/isoamyl alcohol ratio in line 574 and phenol/chloroform ratio in line 582 despite the response saying they are there. Stating product number is insufficient and the authors must take care to allow others to be able to repeat the experiments - however trivial it may seem to the authors - without having to force others to flip through multiple references to find this. "Standard phenol:chloroform extraction" is not descriptive enough since it may only be "standard" to the eyes of the beholder.

And finally, please re-write the sentence "Several potential... to study" (lines 441-442) since it sounds very awkward. One possibility is to simply end the sentence with "Several potential mechanisms are possible" and start the "possibilities" with "For example..." but making this change will have a slight domino effect down this paragraph so please make sure it still flows ok and is grammatically agreeable.

Reviewer #1 (Remarks to the Author):

The authors have made a commendable effort to address the reviewers' comments, especially given the constraints they mention. I think this is an important contribution.

We thank reviewer 1 for her positive comments and helpful suggestions.

Reviewer #2 (Remarks to the Author):

The authors have made a good effort to improve their original manuscript by either toning down insufficiently supported claims or performing additional experiments and analyses. This is highly appreciated, given the difficult working situation. I think this is an important and complete enough study that will promote the interest of the community in the phylogenetic and evolutionary dimensions of post-transcriptional regulation in bacteria. Below are the few remaining editorial points I spotted.

1. L. 138: "Fig. 2h", "Fig. 2i".
2. L. 143: Better would be "Our results suggest that this interaction happens...", since the authors do not show the PrrF1/PrrF2-antR interaction directly.
3. L. 283: Maybe better "... matched the known 5 AAN repeats Hfq motif..." or "...matched a known Hfq motif (5 AAN repeats)..."
4. L. 382: "Fig. 9c".
5. L. 446: "RyhB".

Alexandre Smirnov

We thank reviewer 2 for his positive comments and helpful suggestions. We addressed the 5 remaining editorial points.

Reviewer #3 (Remarks to the Author):

The re-submitted manuscript is a greatly improved version from the previous. There are some minor parts, mostly editorial in nature, that need to be tightened up before being accepted.

We thank reviewer 3 for their positive comments and helpful suggestions.

Borrowing from reviewer 2's comment 15, I do not think changing the error bar from standard deviation to standard error of the mean suddenly makes $n = 2$ statistically significant/acceptable sample size.

We agree that this is not ideal. As stated before, we unfortunately cannot redo this experiment.

Response to 1 (qRT-PCR is indeed featured in the methods, and corresponds to the qRT-PCR experiments

shown in Figure S1.): This is entirely up to you and will not dictate whether the manuscript gets accepted or not, but it's worth thinking about perhaps having a separate M&M section for supplementary materials to de-clutter and abridge the already long article.

As the supplementary methods section would really only include the one RTq-PCR part, we decided to keep it in the main text.

While I agree that the recent advancements of various high-throughput -omics tools are becoming more and more refined, I think it is still important for the authors to describe what their weaknesses/limitations are so that they can lead the readers into understanding why the follow up experiments were designed in certain ways and/or explain why some conclusions can't be made reliably. The authors do a good job addressing our questions as to the limitations of the tools you used (e.g. RIP-Seq) and I would like them to incorporate them more into the manuscript. I think this will only re-inforce the manuscript to an even better quality to be able to discuss the limitations to their advantages when discussing future directions.

We agree and added a paragraph on this point in the discussion (2nd paragraph).

I understand that the authors are trying to re-classify IMHA87 on pseudomonas.com, but if at the time of this manuscript currently being reviewed (and upon publication), it has not occurred yet, I believe you need to at least make a mention in your article that it is identical to AZPAE15042. There may be some people who is unaware of this connection either and it is an advantage to the readers anyway even if this switch occurs relatively soon. If not, you will be able to use this manuscript upon publication to further convince the operators of pseudomonas.com to make the switch. All in all, I think making just one mention somewhere in the main text and/or material and methods will be a big advantage for the authors. I appreciate the information addressed to us reviewers but I'd like it reflected into the manuscript itself.

We agree and added the AZPAE15042 ID in the main text and in the Strain supplementary table.

In response to the author's response (It is often observed that changing the 5'-UTR sequence of an mRNA influences its translation. Indeed, the substitution can also modify the RNA structure, potentially making the transcript more vulnerable/resistant to RNases or interfere with efficiency of ribosome binding, which could explain the different basal activity.): I absolutely agree with what the authors state here. However, it does not make sense at all, given that figure 9b shows an intact native leader sequence of *vfr* seemingly with only the promoter switched from native to Ptac. Therefore, according to this construct design, it should have an entirely native mRNA structure all the way up to 90 bases into the ORF, and so the sequence of mRNA should not be altered as the authors suggest here. I therefore do not think this explanation satisfies my original query.

We might be misunderstanding the reviewer's point here, but the way that we get it, we still believe our answer fits here. The thing is that on the right part of panel 9d (the part with the lower-case "m" mutation) the *vfr* reporter construct is different, and is actually mutated. This is explained in the text and both white bars with lower background expressions (in the "m" part) correspond to this construct where the *vfr* 5'-UTR has been modified at the location of the RhIS binding. We apologize if we are again answering on a different point.

I cannot spot that the authors have included the phenol/chloroform/isoamyl alcohol ratio in line 574 and phenol/chloroform ratio in line 582 despite the response saying they are there. Stating product number is insufficient and the authors must take care to allow others to be able to repeat the experiments - however trivial it may seem to the authors - without having to force others to flip through multiple references to find this. "Standard phenol:chloroform extraction" is not descriptive enough since it may only be "standard" to the eyes of the beholder.

We thank reviewer 3 for insisting on this point as the previous version indeed still lacked important details needed for others to repeat the experiments. We added the ratio information for the specific reagent and additionally described in more details the following steps of the procedure.

And finally, please re-write the sentence "Several potential... to study" (lines 441-442) since it sounds very awkward. One possibility is to simply end the sentence with "Several potential mechanisms are possible" and start the "possibilities" with "For example..." but making this change will have a slight domino effect down this paragraph so please make sure it still flows ok and is grammatically agreeable

The sentences were changed.